

**Title:** Extreme carbon fluxes may result from autochthonous particulate organic
carbon regulated by the interactions between picophytoplankton and heterotrophic
bacteria in river-reservoir systems
**Authors:** Fang Luo [1,2,3], Zhe Li [2,3], Qiong Tang [1,2,3], Yan Xiao [2,3], Lunhui Lu [2,3],
Dianchang Wang [4], Chong Li [4], and Xinghua Wu [4]
**Affiliations:**
[1] Key Laboratory of Hydraulic and Waterway Engineering of the Ministry of
Education, Chongqing Jiaotong University, Chongqing, 400074, China
[2] CAS Key Lab of Reservoir Environment, Chongqing Institute of Green and
Intelligent Technology, Chinese Academy of Sciences, Chongqing, 400714, China
[3] College of Resources and Environment, Chongqing School, University of Chinese
Academy of Sciences, Chongqing, 400714, China
[4] China Three Gorges Corporation, Wuhan, 430010, China
**Correspondence:** Zhe Li (lizhe@cigit.ac.cn)



## Abstract

Freshwater is a significant natural source of atmospheric methane ($CH_4$) and carbon dioxide ($CO_2$) while also receiving significant amounts of particulate organic carbon (POC) from various origins. The variation in carbon ($CH_4$ and $CO_2$) fluxes in freshwater systems is heavily influenced by the sources of POC. The trophic interaction between picophytoplankton (PP) and heterotrophic bacteria (HB) plays a vital role in the carbon cycle within the aquatic system. However, the contributions of different sources of POC to the concentrations and fluxes of $CH_4$ and $CO_2$ are still unclear. Here, we explored the contribution of POC from different sources to extreme carbon emission and the interaction between PP and HB. The evidence from isotope analysis further proved that the extreme carbon fluxes were strongly influenced by autochthonous POC rather than allochthonous POC. Network analysis showed that the positive interaction strength between phytoplankton and bacterioplankton in extreme carbon groups was higher than in normal carbon groups. The results of the structure equation modeling analysis also highlighted that the PP-HB interaction strongly drove the extreme carbon values. This study first introduced the probability statistics method to identify and classify high or low extreme carbon values. These findings also highlight the importance of PP and HB in carbon extreme emissions, and we hope our study can provide an important implication for integrating PP-HB interaction into predicting extreme carbon emissions in the river-reservoir ecosystem.

**Keywords:** Organic carbon; Autochthonous; Allochthonous; Picophytoplankton; Methane emissions



## 1. Introduction

Freshwaters are considered important sources of greenhouse gases (GHGs) to the atmosphere (Bauduin et al., 2024). According to estimates in the Sixth Assessment Report by the Intergovernmental Panel on Climate Change (IPCC), annual global $CH_4$ and $CO_2$ emissions from freshwaters are estimated to be approximately 1.5 Pg $CO_2$ and 159 Tg $CH_4$ (IPCC, 2021), offsetting approximately 25% of the terrestrial carbon sink (Emilson et al., 2018). However, these estimates have a high degree of uncertainty, mainly due to the apparent spatiotemporal heterogeneity and variability of $CH_4$ and $CO_2$ fluxes across the air-water interface. Thus, it is crucial to reduce these uncertainties in emission estimation from a local to a global scale by improving the understanding of fluctuations in $CH_4$ and $CO_2$ concentrations and air-water fluxes.

There have been extensive studies on the cause of significant fluctuations in freshwater $CH_4$ and $CO_2$ fluxes. The hydrological and hydrodynamic conditions, such as river flow, drought, or floods, are significant factors that regulate this variability. Additionally, meteorological factors like short-term heavy precipitation and winds are non-negligible and significant physical factors causing the large variability of freshwater $CH_4$ and $CO_2$ fluxes (Wang et al., 2008). The physical disturbances not only affect the intensity of turbulence mixing at the air-water interface, which changes the rate of air-mass transfer, but also trigger significant input of terrigenous organic carbon (OC) and essential nutrients into freshwater, leading to increased $CH_4$ and $CO_2$ emissions (Liikanen et al., 2002). For example, the decomposition of a significant input of terrigenous OC in the littoral area of freshwater might lead to the ebullition



emission of $CH_4$ in the summer. On the other hand, ecosystem-level events
significantly result in extreme values of $CH_4$ and $CO_2$ fluxes as well. The air-water
$CH_4$ flux was expected to exhibit extremely high values during algal blooms,
concurrently with low levels of surface water $CO_2$ concentrations, leading to an
apparent $CO_2$ sink during the blooming period (Sun et al., 2021). It appeared plausible
that several ecological factors or processes could contribute to the occurrence or
outbreaks of these high or low extremes of $CH_4$ and $CO_2$ concentrations in surface
water and their air-water fluxes. Yet, compared with the physical processes, how
ecological factors or processes could trigger extreme C emissions in freshwaters are
not frequently addressed. New mechanisms are needed to be elucidated.
The minor component of the planktonic communities (Stockner and Antiam,
1986), the picoplankton (defined by a cell size of 0.2-2 μm) (Sieburth et al., 1978),
mainly includes autotrophic picophytoplankton and heterotrophic bacteria (Stockner,
1988). Picophytoplankton are active and critical primary producers in aquatic
ecosystems due to their wide distribution, rapid growth rates, and metabolic
capabilities (Stockner, 1988). In the ocean, picophytoplankton can contribute 50-90%
of primary productivity (Poulton et al., 2006), significantly providing autochthonous
organic carbon to the aquatic ecosystem. Especially during an algal bloom,
small-sized phytoplankton, such as picophytoplankton, can fix more $CO_2$ through
photosynthesis. This is because picophytoplankton have higher growth rates and are
more effective in nutrient and light acquisition than larger phytoplankton (Irion et al.,
2021). Research in the past decade has also shed new light on a large proportion of



tiny picophytoplankton to carbon export, especially in oligotrophic oceans
(Richardson, 2019). On the other hand, heterotrophic bacteria decompose organic
carbon, transferring different sources of OC into $CH_4$ or $CO_2$ (Guillemette et al.,
2013). It was reported that heterotrophic bacteria can consume 20-60% of the total
primary production (Williams, 1981). The effects of heterotrophic bacteria on $CH_4$ or
$CO_2$ emissions are strongly dependent on the decomposition of organic carbon with
different bioavailability (Grasset et al., 2018). Therefore, as an important part of the
planktonic communities, picophytoplankton and heterotrophic bacteria undoubtedly
are essential components for understanding the carbon cycle in the aquatic ecosystem.
Interactions between picophytoplankton and heterotrophic bacteria are critical
for exploring the possible mechanisms that regulate $CH_4$ and $CO_2$ dynamics in aquatic
ecosystems. Picophytoplankton and heterotrophic bacteria do not exist in isolation
(Faust and Raes, 2012), and there are complex ecological interactions between them,
which span mutualism, commensalism, parasitism, and competition (Seymour et al.,
2017). In brief, the relationship between picophytoplankton and heterotrophic bacteria
is based on resource provision and can be either cooperative (exchange of resources)
or competitive (competition for resources) (Amin et al., 2012). Mutualism, a win-win
relationship, is found to be the primary relationship between these two
microorganisms (Zhang et al., 2021). One example for mutualism is cross-feeding, in
which two species exchange metabolic products to facilitate the growth of both
(Woyke et al., 2006). For instance, heterotrophic bacteria directly obtain a large
proportion of picophytoplankton-derived organic carbon to meet their carbon demand



(Zhou et al., 2022). Autrophic picophytoplankton can utilize vitamins and
micronutrients (that is, iron, copper, etc.) released by heterotrophic bacteria (Zhou et
al., 2022; Durham et al., 2015). There are two possible mechanisms by which
picophytoplankton-heterotrophic bacteria interactions affect $CH_4$ and $CO_2$ flux. First,
the cooperative relationship between picophytoplankton and heterotrophic bacteria
produces strong coupling and positive feedback between these two organisms,
increasing microbial metabolic efficiency and full utilization of OC (Coyte et al.,
2015). Second, "physical interactions" between picophytoplankton and heterotrophic
bacteria in an extracellular microenvironment (that is, "phycosphere") mediate the
level of aggregation of picophytoplankton biomass, which manipulates downward C
flux (Seymour et al., 2017). Despite their small size, more than 40% of
*Synechococcus* cells were found to be conjoint with heterotrophic bacteria ("physical
interaction") in situ observation (Malfatti and Azam, 2009). Such an increase in
picophytoplankton and heterotrophic bacteria cell aggregation mediated by
interactions between picophytoplankton and heterotrophic bacteria, especially during
the algal blooms, would lead to an increase in the carbon flux exported to the bottom
water column (Gärdes et al., 2011), thus offering more substrate for $CH_4$ production.
In recent years, the influence of picophytoplankton and heterotrophic bacteria on the
biogeochemical cycle of carbon has been widely discussed in marine ecosystems
(Zhou et al., 2022). Little is known about the dynamics of $CH_4$ and $CO_2$ production
and emissions driven by the interaction of these "specific participants" in freshwaters.



River damming disrupts the natural connectivity of rivers and causes a shift in
the aquatic system from lotic to lentic type along the longitudinal gradients towards
the dam site (Baxter, 1977). This change significantly affects the river's flow by
reducing speed, prolonging the hydraulic retention time, and interrupting sediment
movement (Maavara et al., 2020). Reservoirs receive a higher input of terrigenous
organic carbon than natural lakes due to their comparably lower ratio of watershed
area to surface area and higher shoreline development (Thornton et al., 1990). Over
the past two decades, there has been increasing concern about the excessive carbon
emissions from reservoirs. This is significant for global carbon biogeochemical cycles
and has implications for the hydropower industry. However, organic carbon sources
contributing to carbon emission, especially extremes, have yet to be well explored.
This lack of understanding hinders the accurate prediction of reservoir carbon
emissions in various scenarios.
Although extremely high or low $CH_4$ and $CO_2$ concentrations in surface water or
their air-water fluxes were not frequently detected, it was assumed that the extremes
or normal status of C fluxes could represent the distinctive ecosystem-level state and
the biogeochemical cycling. Thus, processes and mechanisms of carbon cycling in the
river-reservoir system could be further explored through the categorization of extreme
or normal status of $CH_4$ and $CO_2$ concentrations or air-water fluxes. Therefore, we
hypothesized that (1) autochthonous organic carbon (OC) in river-reservoir systems
greatly contributes to the occurrences of extreme values of $CH_4$ and $CO_2$
concentrations; (2) terrigenous OC contributes to the normal values of $CH_4$ and $CO_2$



concentrations; and (3) The interaction of autotrophic picophytoplankton (PP) and
heterotrophic bacteria (HB) could be intensified with an increase in trophic state, thus
promoting the production of extreme values of $CH_4$ and $CO_2$.

To test the hypothesis, we first identified and classified extremely high or low

values of $CH_4$ and $CO_2$ concentrations and their air-water fluxes across different types
of reservoirs in the upper Yangtze River basin in China. Then, we investigated
variations and interactions of picophytoplankton and heterotrophic bacteria, together
with environmental parameters and stable isotopic evidence. Building on our
sampling campaign (Tang et al., 2023), we restructured the information and conducted
new analyses. This study enhances prior research through two key contributions:

1) We categorized the concentrations and fluxes of $CH_4$ and $CO_2$ from the

previous datasets into two groups: extreme and normal. This classification is based on
the probability of occurrence and offers new insights into the mechanisms that
regulate these gases.

2) We included new data from pico-phytoplankton and heterotrophic bacteria.

This information may provide fresh evidence regarding the interactions between
phytoplankton and bacteria that contribute to $CH_4$ emissions in the extreme group.

Hopefully, our study will determine the role of picophytoplankton and

heterotrophic bacteria in extreme carbon emissions, which would yield new insights
into extreme carbon emissions with the two tiny planktonic communities in
river-reservoir systems.



**2. Materials and methods**

*2.1. Study sites and sample collection*

Five reservoirs in the upper Yangtze River basin were selected (Fig. 1), including Xiluodu (XLD), Xiangjiaba (XJB), Shizitan Reservoir (SZT), Xiaoba II (XB II) and Three Gorges Reservoir (TGR). Among these five reservoirs, XLD, XJB, and TGR are located on the main stem of the Yangtze River, which are large river-valley dammed cascade reservoirs and mainly serve as hydropower generation and flood control. The SZT (also known as Changshou Lake) is located on the Longxi River, a tributary of the Yangtze River, and functions as tourism now. XB II (with a total capacity of 11300 m³) is a small reservoir only for drinking water supply located on a tertiary tributary named Ganxi Gulley of the Yangtze River. The geographical and project information of these selected reservoirs is described in Table S1.

Sampling campaigns were conducted in May, July, and November 2019 to obtain a representative dataset containing different seasons. Twenty-six sampling sites were set in the five selected reservoirs, covering the riverine zone, transitional zone, and lacustrine zone of each reservoir (Fig. 1; Table S2). Water samples for cell enumeration of picophytoplankton and heterotrophic bacteria were filtered through a 50-μm nylon sieve. The filtered samples were immediately fixed by glutaraldehyde solution and kept at -80 °C in the laboratory until analysis for flow cytometry analysis. Water samples for bioinformatic analysis of phytoplankton and bacterioplankton were filtered through 0.22 μm Millipore cellulose filters (Milford, USA). The filtered membranes were then kept at -86°C until DNA extraction. The remaining water



samples for analysis of environmental parameters were pretreated according to
standard methods.
*2.2. Physicochemical parameters*
Water temperature (WT), dissolved oxygen (DO), and pH were measured on-site
with a multiparameter sonde (YSI®EXO2, USA). The concentrations of chlorophyll a
(Chl-a) and different forms of nitrogen and phosphorus in water were measured
according to the Monitoring Analysis Method of Water and Wastewater (SEPA, 2002)
using a UV-visible spectrophotometer (Shimadzu® UV2700i, Japan).
Concentrations of $CH_4$ and $CO_2$ in the water phase were measured with the
headspace equilibration method (Goldenfum, 2010). In brief, a water sample (200 mL)
was gently collected using a polypropylene syringe equipped with a three-way valve.
100 mL $N_2$ (99.999%) gas was added to the syringe to create a headspace. After 3 min
of vigorous shaking, the equilibrated headspace gas was injected into a pre-evacuated
airbag (Delin® 300 mL, Dalian) for storage until measurement. Gas samples were then
analyzed using a stable isotope analyzer (Picarro® G2201-i, USA). The $CH_4$ and $CO_2$
emission fluxes at the water-air interface were estimated by the thin boundary layer
method (Goldenfum, 2010). All measurements were performed in triplicate for quality
assurance.
The frozen filtered POC membranes were dried at 65 ℃ for 48 h, fumigated with
HCl (12 M) for 12 h to remove particulate inorganic carbon, and wrapped in a tin boat.
The wrapped filtered POC membranes were used to analyze the concentrations of
POC and PON using a stable isotope mass spectrometer coupled with an elemental



analyzer (Thermo Fisher Scientific® Flash HT-Delta V Advantage, USA).
*2.3. Analysis of microbial communities*
Picophytoplankton and heterotrophic bacteria abundance (< 2 μm) in collected
samples was determined using a flow cytometer (Beckman Coulter® CytoFLEX,
USA). Based on previous methods (Lu et al., 2018; Yang et al., 2019), we pretreated
flow cytometry samples and set the discriminator of flow cytometry. Briefly, for
picophytoplankton enumeration, 1 mL of unstained sample was taken for analysis.
The discriminator was set on two red and one orange fluorescence, respectively. For
heterotrophic bacteria enumeration, 20 μL samples were diluted into 1 mL with
sterilized water and stained with SYBR Green I (Molecular Probes, USA) for 15 min
at room temperature in the dark. The discriminator was set on red and green
fluorescence, respectively. Detailed flow cytometry analysis of picophytoplankton and
heterotrophic bacteria was described in supplementary method S1.
Genomic DNA was extracted and duplicated from the filters with a DNA
isolation kit (Mo Bio laboratories® FastDNA SPIN kit, USA) according to the
manufacturer's instructions. The duplicate DNA extracts were mixed for the following
PCR amplification. The primers used for the phytoplankton 23S rRNA gene were
A23SrVF2 and A23SrVR2 (Yoon et al., 2016). The primers used for bacterioplankton
16S rRNA gene were 338F and 806R (Ding et al., 2020). Amplicons were purified
with an AxyPrep DNA Gel Extraction Kit (Axygen Biosciences, USA). The PCR
products of each sample were sequenced on the Illumina MiSeq platform at Majorbio
Bio-Pharm Technology, Co., Ltd. (Shanghai, China). DNA extraction, PCR



amplification, and high-throughput sequencing were discussed in detail in the
supplementary method S2.
*2.4. Stable carbon isotopic analysis*

$\delta^{13}C$ of $CH_4$ and $CO_2$ in the water phase were analyzed using a stable isotope

analyzer (Picarro® G2201-i, USA). Stable isotopes of POC and PON were measured
using a stable isotope mass spectrometer coupled with an elemental analyzer (Thermo
Fisher Scientific® Flash HT-Delta V Advantage, USA). $\delta^{13}C$ and $\delta^{15}N$ of POM were
used to determine the contributions of different sources of POC. Stable isotope values
of endmembers (C3 plant, C4 plant, coastal soil, and plankton) were summarized in
Table S3. DOC concentration and $\delta^{13}$-DOC were analyzed by a total organic carbon
analyzer-stable isotope mass spectrometer (Elementar® vario cube TOC-isoprime100,
Germany).

In this study, we applied isotopic analysis to specifically explore the impact of

organic carbon from autochthonous sources on extreme values of $CH_4$ and $CO_2$
production. We focused on the two main mechanisms affecting $\delta^{13}C$-$CH_4$ and
$\delta^{13}C$-$CO_2$: (i) physical proces and (ii) biological processes (Han et al., 2018). Our
analysis concentrated on the variability of $CH_4$ and $CO_2$ in the water column, rather
than at the air-water interface, using isotope data. We made the assumption that the
fractionation effect induced by $CH_4$ and $CO_2$ exchange at the air-water interface, such
as dissolution and emission, can be ignored.





*2.5. Identification of extreme and normal levels of CH$_4$ and CO$_2$ concentrations and*
*fluxes*
Pearson type III probability distribution curve (Hosking and Wallis, 1997), a
widely used probability distribution function in hydrology and meteorological
statistics (Sun and Qin, 1989), such as frequency analysis of extreme hydrological
events, risk assessment of extreme climate, etc. (Raynal Villaseñor, 2021). Here, we
employed Pearson type III probability distribution to determine the extreme and
normal values of CH$_4$ and CO$_2$ concentrations and fluxes. This method allowed us to
calculate the threshold values for extreme values of CH$_4$ and CO$_2$ concentrations and
fluxes at 10% and 90% probabilities (Ding and Jiang, 2009). Based on these threshold
values, we divided the dataset into three groups, i.e. extremely high (Ext_h), normal
(nor), and extremely low (Ext_l). The supplementary material method S3 provided
threshold values and sample numbers for the above three groups of CH$_4$ and CO$_2$
concentrations and fluxes, respectively.
*2.6. Statistical analysis*
Originpro (OriginLab®, USA, education version) was used for graphing. Data
analyses were performed by SPSS (IBM, USA). Differences among groups were
considered to be statistically significant if $p < 0.05$. Multiple linear regression analysis
was conducted to determine whether the main predictor of carbon extreme values was
POCauto or POCallo.
The trophic state of each sampling site was evaluated based on the trophic level
index (TLI) (Tang et al., 2023). Alpha diversity (Shannon-Wiener and Chao1 index)



was calculated by using the *vegan* package. The contributions of different sources to
POC were estimated using Bayesian stable isotope mixing models with the *simmr*
package (Parnell et al., 2013). The co-occurrence networks of phytoplankton and
bacterioplankton were constructed by 16S rRNA and 23S rRNA microbial ASVs
using *igraph* package. The main predictors for the abundance of picophytoplankton
and heterotrophic bacteria were identified by random forest (RF) analysis with the
*randomForest* package. Structure equation modeling (SEM) analysis was
implemented by the *lavaan* package to explore the relationships among all variables
for both the extreme and normal groups. All these methods mentioned above were
described in detail in the supplementary material methods. The setting of the sampling
campaign and analysis of water samples were described in our prior research (Tang et
al., 2023). All data analyzed in this study are sourced from the same dataset (Tang et
al., 2023).
**3. Results**
*3.1. Identification of extreme and normal values of carbon concentrations and fluxes*
The extreme and normal values for $CH_4$ and $CO_2$ concentrations and fluxes are
shown in Table 1. The mean $CH_4$ concentration ($CCH_4$) in the normal and extremely
high groups were 0.03±0.00 and 0.19±0.02 μmol·$L^{-1}$, respectively. The mean $CH_4$ flux
($FCH_4$) in the extremely low, normal, and extremely high groups were 0.01±0.00,
0.10±0.01, and 0.61±0.14 mmol·$m^{-2}$·$d^{-1}$, respectively. The mean $CCO_2$ in the
extremely high and normal groups were 6.23 and 3.51 times higher than in the
extremely low group, respectively. The mean $FCO_2$ in the extremely low, normal, and



extremely high groups were $0.05\pm0.83$, $25.72\pm1.16$ and $62.71\pm8.56$ mmol·m$^{-2}$·d$^{-1}$,
respectively. Differences in $CH_4$ and $CO_2$ concentrations and fluxes between extreme
and normal groups were significant ($p < 0.001$). Since the $CH_4$ concentration in the
whole dataset was higher than the threshold value for the extremely low $CH_4$
concentration ($0.004$ μmol·L$^{-1}$), the number of samples corresponding to the
extremely low group of $CH_4$ concentration was 0. Thus, the extremely low group
(Ext_l) of $CH_4$ concentration ($CCH_4$) was nonexistent.
The trophic levels were noticeably different between extreme and normal groups
of $CH_4$ and $CO_2$ concentrations and fluxes (Table 2). The TLI values in the extremely
high group of $CCH_4$ and $FCH_4$ were higher than 46.27, which belonged to the
eutrophic state. The TLI values fluctuated within 38.91-46.27 in the normal group of
$CCH_4$ and $FCH_4$, indicating the water in the normal group was mesotrophic. With the
increase of trophic state, $CCH_4$ and $FCH_4$ exhibited a increasing trend, and $CCH_4$ and
$FCH_4$ ranged from 0.07 to 0.20 μmol·L$^{-1}$ and 0.06 to 0.18 mmol·m$^{-2}$·d$^{-1}$, respectively
(Fig. S3). Both extremely high and extremely low groups of $CCO_2$ and $FCO_2$ were
eutrophic, but the normal group was mesotrophic (Table 2). $CCO_2$ and $FCO_2$
decreased from oligotrophic state to eutrophic state, and $CCO_2$ and $FCO_2$ ranged from
44 to 43 μmol·L$^{-1}$ and 24 to 23 mmol·m$^{-2}$·d$^{-1}$, respectively (Fig. S3).
Ternary plots showed that the extremely high values of $CCH_4$ mainly occurred in
July, with a relative percentage up to 56%, than in other months (relative percentage
of 44% in May and 0% in November), while most of the normal values of $CCH_4$
appeared in November, accounting for 38% of total normal groups (Fig. 2A). We



further observed that the extremely low, normal, and extremely high values of $FCH_4$
mainly occurred in November, May, and July, respectively. On the contrary, the
extremely low, normal, and extremely high values of $CCO_2$ and $FCO_2$ mainly
occurred in July, May, and November, respectively. These results all exhibited that
extreme level of carbon concentrations and fluxes (extremely high values for $CH_4$ and
extremely low values for $CO_2$) mostly appeared in July, which supports the inference
that cell aggregation mediated by the PP-HB interaction drives the extreme values of
$CH_4$ and $CO_2$, especially during the summer blooming period.

As Fig. 2B shows, most environmental parameters differed significantly among

extremely low, normal, and extremely high groups of $CH_4$ and $CO_2$ concentrations and
fluxes, with $p < 0.05$. Except for WT and POC, other environmental factors did not
show significant differences among the extreme and normal groups of $CCH_4$ and
$FCH_4$. The mean WT and POC exhibited an increasing trend from the extremely low
to extremely high group of $CCH_4$ and $FCH_4$ ($p < 0.05$).

Results also showed that the mean $NO_3^--N$ concentration in the normal group of

$CCO_2$ and $FCO_2$ was significantly higher than that in the extremely low group of
$CCO_2$ and $FCO_2$ (Fig. 2B; $p < 0.01$). An increasing trend in mean SRP concentration
was observed sequentially from the extremely low to the extremely high group of
$CCO_2$ and $FCO_2$, and the mean SRP was lowest in the extremely low group (0.02
mg·L$^{-1}$ for $CCO_2$; 0.02 mg·L$^{-1}$ for $FCO_2$) and highest in the extremely high group
(0.07 for $CCO_2$ mg·L$^{-1}$; 0.07 mg·L$^{-1}$ for $FCO_2$); conversely, the mean DO, WT, pH
were highest in the extremely low group and lowest in the extremely high group of



$CCO_2$ and $FCO_2$. Moreover, POC and DOC in the extreme groups (high or low) of
$CCO_2$ were higher than those in the normal group.
*3.2. Contributions of autochthonous and allochthonous POC*

The concentrations of autochthonous POC (POCauto) and allochthonous POC

(POCallo) were noticeably different among the extreme and normal groups of $CH_4$
($CCH_4$) and $CO_2$ concentrations ($CCO_2$) (Fig. S4). The POCauto and POCallo
concentrations in whole dataset respectively ranged from 0.004 to 0.859 $mg·L^{-1}$ and
0.05 to 1.77 $mg·L^{-1}$, and were positively correlated with TLI (Figs. S4 and S5). In the
extreme and normal groups of $CCH_4$, mean concentrations of POCauto and POCallo
in the extremely high group (0.22±0.08 $mg·L^{-1}$ for POCauto; 0.46±0.14 $mg·L^{-1}$ for
POCallo) were significantly higher than those in the normal groups (0.07±0.02 $mg·L^{-1}$
for POCauto; 0.30±0.05 $mg·L^{-1}$ for POCallo) (Fig. S4). In the extreme and normal
groups of $CCO_2$, mean POCauto and POCallo concentrations in the extremely low
groups were significantly higher than those in the normal groups (Fig. S4).

Furthermore, POCauto and POCallo were both positively correlated with the

$CCH_4$, respectively (Fig. 3A). In the extremely high group of $CCH_4$, we observed a
greater slope value between POCauto and $CCH_4$ than that between POCallo and
$CCH_4$. However, in the normal group of $CCH_4$, the slope value between POCallo and
$CCH_4$ was higher than that between POCauto and $CCH_4$. Statistical analysis showed
that $CCO_2$ was significantly positively correlated with POCauto, but the relationships
between $CCO_2$ and POCallo were not significant (Fig. 3B). The slope values between
POCauto and $CCO_2$ were greater than those between POCallo and $CCO_2$ among the



three groups. The increase in the autochthonous POC contributed to a higher $C_{CH_4}$
(Fig. 3C) and lower $C_{CO_2}$ (Fig. 3D), respectively.
*3.3. Picophytoplankton and heterotrophic bacteria abundance across extreme and*
*normal groups*
The abundances of picophytoplankton (PP) and heterotrophic bacteria (HB)
varied with extreme and normal groups of $CO_2$ and $CH_4$ concentrations (Fig. 4). The
abundance of PP and HB respectively ranged from $0.01 \times 10^5$ to $5.66 \times 10^5$ cells·mL$^{-1}$
and $0.26 \times 10^5$ to $8.90 \times 10^5$ cells·mL$^{-1}$ in the whole dataset (Fig. 4). The difference in PP
or HB abundance between extremely high (Ext_h) and normal groups (Nor) of $CH_4$
concentration was not significant (Fig. 4A). The HB abundance was higher than that
of PP in the high (Ext_h) and normal groups (Ext_h) of $CH_4$ concentration (Fig. 4B).
The differences in HB abundance were not significant in three extreme and normal
groups of $C_{CO_2}$ (Fig. 4C). The mean PP abundance in the extremely low group (Ext_l)
of $CO_2$ concentration was significantly higher than those in the normal (Nor) and
extremely high groups (Ext_h) of $CO_2$ concentration. The HB abundance was higher
than that of PP in the extremely high (Ext_h) and normal groups (Nor) of $CO_2$
concentration except in the extremely low group (Ext_l) of $CO_2$ concentration (Fig.
4D).
We used random forest (RF) modeling (Fig. 5) to identify the main
environmental predictors of picophytoplankton and heterotrophic bacteria abundance.
The main environmental predictors influencing PP abundance were TP, SRP, DIN,
NO$_3^-$-N, Chl-a, DO, WT, pH, POC, and DOC ($p < 0.05$) (Fig. 5A). The main



environmental predictors also had a certain influence on the HB abundance ($p < 0.05$)
(Fig. 5B). The variance explanation of environmental factors for the abundance of PP
($R^2$=0.82) was higher than that of HB ($R^2$=0.39).
*3.4. Interactions between phytoplankton and bacterioplankton communities*

The co-occurrence patterns of phytoplankton and bacterioplankton in the

extreme and normal groups of $CH_4$ and $CO_2$ concentrations were determined using
network analysis (Fig. 6; Table 3). Overall, the number of nodes ranged from 101 to
184 in all five interaction networks. Most networks consisted of more than 50%
positive edges, except in networks for the normal group of $CO_2$ concentration. In
addition, topological properties of the co-occurrence network in normal groups of
$CH_4$ and $CO_2$, such as modularity, were higher than those in the extreme groups. In
contrast, the average degree showed an exactly opposite trend. The number of
phytoplankton-bacterioplankton links decreased from 977 in the extremely high group
to 104 in the normal group of $CH_4$ concentration (Fig. 6A and B). Similarly, the
number of phytoplankton-bacterioplankton links in the normal group was also
significantly lower than both in the extremely high and extremely low groups of
$CO_2$ concentration (Fig. 6C-E). Compared with communities in normal groups,
communities in the extreme groups exhibited a higher interaction strength (that is,
links) between phytoplankton and bacterioplankton.
*3.5. Influential pathways of planktonic communities on $CH_4$ and $CO_2$ concentrations*

We conducted structural equation modeling (SEM) to understand the direct and



indirect relationships between microbial variables and POC with $CH_4$ and $CO_2$
concentrations for extreme and normal groups (Fig. 7). Our SEMs explained 96 and
21% of the variance in $CH_4$ concentrations in the extremely high and normal groups,
respectively (Fig. 7A and B). In the extremely high group of $CH_4$ concentration, $CH_4$
concentration was directly influenced by picophytoplankton (PP), autochthonous POC
(POCauto), and allochthonous POC (POCallo) with path coefficients of 0.77, 0.35
and -0.26, respectively (Fig. 7A). Picophytoplankton could also indirectly affect the
extremely high $CH_4$ concentration by influencing POCauto and POCallo. POCauto
and POCallo were positively and negatively correlated with the extremely high $CH_4$
concentration ($p < 0.01$ and $p < 0.001$, respectively). However, HB didn't show a
significant influence on the extremely high $CH_4$ concentration. In the normal group
(Fig. 7B), both POCauto and POCallo were positively correlated with the $CH_4$
concentration ($p < 0.001$ and $p < 0.01$, respectively), and the path coefficient between
POCallo and $CH_4$ concentration was greater than that between POCauto and $CH_4$
concentration. However, Chl-a had a negatively direct effect on the normal value of
$CH_4$ concentration with a negative path coefficient of 0.58 ($p < 0.01$).

The selected variables explained 85, 26, and 96% of the variance in $CO_2$

concentrations in the three groups, respectively (Fig. 7C-E). In the extremely high
group of $CO_2$ concentration, POCauto, POCallo, and HB were not correlated with the
$CO_2$ concentration, while PP significantly affected the extremely high $CO_2$
concentration ($p < 0.001$) (Fig. 7C). In the normal group of $CO_2$ concentration, only
POCallo had a positively direct impact on the normal value of $CO_2$ concentration



($p$<001) (Fig. 7D). There was only one indirect path in which PP affected the normal
value of $CO_2$ concentration by influencing POCallo. In the extremely low group of
$CO_2$ concentration, except for POCallo, PP, POCauto, and HB showed a significant
direct impact on the extremely low value of $CO_2$ concentration (Fig. 7E, $p$ < 0.001,
respectively). PP significantly affected POCauto ($p$ < 0.01) and ultimately affected the
extremely low value of $CO_2$ concentration.
**4. Discussion**
*4.1. Contributions of POC from different sources to $CH_4$ and $CO_2$*
$CH_4$ and $CO_2$ are the dominant gaseous end products of organic carbon (OC)
decomposition (Yvon-Durocher et al., 2011), but the potential for $CH_4$ and $CO_2$
production and emissions differs between different sources of POC (Berberich et al.,
2020). In the present study, we hypothesized that (1) the extreme values of $CH_4$
($CCH_4$) and $CO_2$ concentrations ($CCO_2$) will mainly be fueled by autochthonous POC
(POCauto), (2) the normal values of $CH_4$ and $CO_2$ concentrations will mainly be
stimulated by allochthonous POC (POCallo).
Previous studies proved that autochthonous OC will decompose faster than
allochthonous OC, thus sustaining higher $CO_2$ and $CH_4$ production rates than
allochthonous OC (Grasset et al., 2018). This evidence probably supports the view
that autochthonous POC supports short-term carbon production and emissions. Our
study found steeper slopes between carbon ($CH_4$ and $CO_2$) concentrations and
autochthonous POC than those between carbon concentrations and allochthonous
POC in the extreme groups of $CH_4$ and $CO_2$ (Fig. 3A and B). There was more positive



correlation between autochthonous POC and the extreme values of $CH_4$ and $CO_2$
concentrations than those between allochthonous POC and the extreme values of
carbon (Table 4), indicating the main contribution of autochthonous POC to the
extreme values of $CH_4$ and $CO_2$ concentrations. The evidence from isotope analysis
further proved and highlighted that the extreme values of $CH_4$ and $CO_2$ concentrations
were strongly influenced by autochthonous POC (Fig. 8). Photosynthesis and
decomposition were the important biological processes controlling concentrations and
isotope values of OC and carbon. The input of OC to reservoirs is a complex mixture
of autochthonous OC (i.e., OC derived from aquatic primary production) and
allochthonous OC (i.e., OC derived from terrigenous input) (Chen et al., 2021). In the
extreme group of $CH_4$ concentration, POC in the surface water was mainly from
aquatic plankton, and the average $\delta^{13}C$-POC was approximately -26.34‰ (Table S3).
Phytoplankton provides a large amount of autochthonous POC through
photosynthesis, and picophytoplankton was therefore significantly correlated with
POC ($p < 0.01$, Fig. 8A). However, in summer, phytoplankton may assimilate $HCO_3^-$
as an inorganic C source under $CO_2$ limitation during intense photosynthesis, which
weakens the discrimination of $^{13}C$ and enriches organic matter with $^{13}C$ (Fogel and
Cifuentes, 1993). The above reasoning explained why the $^{13}C$ enrichment in the
surface water DOC with the increase of the Chl-a ($p < 0.01$, Fig. 8B). A positive
correlation between $\delta^{13}C$-POC and $CH_4$ concentration in the extremely high group of
$CH_4$ ($p < 0.05$, Fig. 8C) indicated that the extreme value of $CH_4$ was influenced by the
decomposition of autochthonous POC. This is probably because the decomposition of





OC from phytoplankton preferentially releases $^{12}$C and leaves the residual OC
enriched in $^{13}$C (van Breugel et al., 2005). High productivity may also convert lakes
or reservoirs from a $CO_2$ source to a sink (Balmer and Downing, 2011). The
$\delta^{13}$C-DOC showed a negative correlation with $\delta^{13}$C-$CO_2$ ($p < 0.01$, Fig. 8D), which
could be explained by an increase in phytoplankton photosynthesis (Fig. S6). These
results supported the first hypothesis above.

Nevertheless, because allochthonous POC accounts for large proportions of

aquatic ecosystems, allochthonous POC can support long-term $CCH_4$ accumulation
and emissions (Berberich et al., 2020). Correspondingly, we found steeper slopes
between $CCH_4$ and allochthonous POC than between $CCH_4$ and autochthonous POC
in the normal group of $CCH_4$ (Fig. 3A). The results of SEM also highlighted that the
normal values of $CCH_4$ and $CCO_2$ were positively influenced by allochthonous POC
(Fig. 7B and D). These results supported our second hypothesis above.

There are two main reasons why POC from different sources have varying

impacts on carbon production and emissions. First, chemical structure. Autochthonous
OC, mainly composed of protein and aliphatic compounds, has relatively simple
chemical structure (Kendall et al., 2001). Allochthonous POC, mainly composed of
cellulose, has relatively complex chemical structure (Sondergaard and Middelboe,
1995). Second, biological availability. Complete degradation of allochthonous POC
requires many species with different degradation capabilities, and therefore, the
degradation of allochthonous POC is relatively slow, and the yield of carbon is stable
(Grasset et al., 2018). In contrast, autochthonous POC, such as algal biomass, a labile



carbon source, is easily available for microorganisms (Berberich et al., 2020), thus
making the decomposition of autochthonous POC relatively rapid and the yield of
carbon production variable. In a word, autochthonous POC can stimulate the
production of extreme values of $CH_4$ and $CO_2$ in the short term (low probability of
occurrence), while allochthonous POC can maintain the normal production of $CH_4$
and $CO_2$ in the long term (high probability of occurrence).
*4.2. Response of picophytoplankton and heterotrophic bacteria to trophic state*
Compared to oligotrophic and mesotrophic states, the eutrophic state can provide
more resources for phytoplankton and bacterioplankton, thus reducing resource
competition between different species (Tang et al., 2023). In this study, we found that
the abundances of both picophytoplankton and heterotrophic bacteria in the extreme
groups of CCH4_Ext_h and CCO2_Ext_l (eutrophic state) were slightly higher than
those in the normal groups of $CH_4$ and $CO_2$ concentrations (mesotrophic state) (Fig. 4;
Table 2; Fig. S7). Recent studies reported that the environments with sufficient
resources will reduce niche overlap and enhance coexistence (Pastore et al., 2021).
Niche overlap reflects the degree to which species share the factor controlling their
growth (such as resources) (Pastore et al., 2021), and low niche overlap is mainly
considered weaker competition (Clavel et al., 2011). Our results showed a decrease in
the niche overlap of phytoplankton and bacterioplankton from oligotrophic state to
eutrophic state (Fig. S8), which agrees with Clavel et al. (2011). Furthermore, in
freshwater ecosystems, environmental conditions determine community diversity
(Meng et al., 2020). Alpha diversity is considered positively dependent on



environmental filters (Stefanidou et al., 2020). The increase in the alpha diversity
index was mainly considered as a signal of improved trophic state (Arab et al., 2019).
In our study, we found a decrease in alpha diversity of both phytoplankton and
bacterioplankton from oligotrophic state to eutrophic state (Fig. S9), which agrees
with Meng et al. (2020).
Bacterial taxa tended to be divided into r-strategists and K-strategists according
to the preference of different OC decomposition (Li et al., 2021). Previous studies
reported that the fast-growing bacterial taxa (r-strategists) prefer a environment
enriched with labile C, while the slow-growing bacterial taxa (K-strategists) favor a
nutrient-poor environment (Dai et al., 2022). In this study, an increase in the
proportion of r-strategists to K-strategists with trophic states (Fig. S10) suggested that
high eutrophic state provided more autochthonous POC (Fig. S5), promoting the
growth of r-strategists. In aquatic ecosystems, picophytoplankton and heterotrophic
bacteria have tiny sizes and rapid growth rates and can be characterized as typical
competitors (r-strategists). Phytoplankton communities have simplex richness
predominated by competitors in eutrophic lakes and rivers (Raffoul et al., 2020). In
parallel, prior studies reported that picophytoplankton (PP) contributes 50-90% of
primary productivity (Poulton et al., 2006), and HB consumes 20-60% of the total
primary production (Williams, 1981).
Types of interactions between two species include positive relationships (such as
mutualism and commensalism) and negative relationships (such as competition and
amensalism) (Faust and Raes, 2012). Our results showed that in co-occurrence



networks, the eutrophic communities displayed the highest positive interaction
strength between phytoplankton and bacterioplankton compared with the oligotrophic
and mesotrophic communities (Figs. 6 and S11). Such an increase supported the
inference that the eutrophic state increased the interaction strength between PP and
HB. Increased cell density of PP in the eutrophic state (Fig. S7) would reduce the
distance between PP and HB (Petrou, 2023), and therefore increase the encounter
rates and interactions in the extracellular microenvironment (Christie-Oleza et al.,

2017).

*4.3. Extreme and normal patterns of carbon emissions driven by picophytoplankton*
*and heterotrophic bacteria*
Although some extreme $CO_2$ and $CH_4$ air-water fluxes could be induced by
short-term physical processes, the short-term physical processes were not the focus of
our study. We found that water temperature showed no difference between normal and
extreme groups of $CO_2$ fluxes (Fig. 2B). Yet, the picophytoplankton abundance in the
extremely low group of $CO_2$ concentration was significantly higher than that in the
normal group of $CO_2$ concentration (Fig. 4C). These results suggested that the
extreme and normal values of $CH_4$ and $CO_2$ were probably influenced by ecosystem
response (such as microbial community composition or abundance variation), rather
than physical factors (such as temperature and wind).
In aquatic systems, the growth and functions of microorganisms are influenced
mainly by the quantity and quality of substrates (Yang et al., 2023) entering the water.
Microbial communities, therefore, structure their responses to OC using different life



strategies (Delgado-Baquerizo et al., 2016), which impact water C dynamics. Previous
studies reported that microbial community composition is correlated with substrate
utilization strategy (Schutter and Dick, 2001). The response of microorganisms to OC
from different sources is different on a time scale. Based on the growth rate and
effectiveness of C utilization, microbial communities can be classified into two
ecological functional groups, r- and K-selected species. Picophytoplankton and
heterotrophic bacteria, as r-strategists, have a fast growth rate and a rapid response to
labile C (Li et al., 2021). In contrast, K-selected species are slow-growing, decompose
recalcitrant C more efficiently, and respond slowly to OC inputs. This could explain
why the extreme values of $CH_4$ and $CO_2$ concentrations were positively influenced by
picophytoplankton (Fig. 7). Hence, picophytoplankton and heterotrophic bacteria play
essential roles in maintaining short-term extreme carbon emissions.

Our study found a significant positive correlation between network degree

(interaction strength) and $CH_4$ concentrations in the eutrophic state (Fig. S12). Such a
positive correlation corresponds to the third hypothesis that increased interaction
between picophytoplankton and heterotrophic bacteria promoted the extreme values
of carbon. This is mainly because positive interaction (i.e., cooperation) produces
strong coupling and positive feedback between PP and HB (Coyte et al., 2015),
increasing microbial metabolic efficiency and full utilization of OC.
Picophytoplankton and heterotrophic bacteria have numerous enzymes for
depolymerizing fresh labile C (such as autochthonous OC) (Li et al., 2021) and
typically flourish in environments enriched in unstable C.



Extremely values of $CH_4$ and $CO_2$ concentrations and fluxes were found in July
(Fig. 2A). During the algal blooming period, increased cell density of PP (Fig. S7)
wound enhance the possibility of "physical interaction" between phytoplankton and
heterotrophic bacteria (Christie-Oleza et al., 2017). The increase in PP-HB interaction
facilitates cell aggregation, increasing carbon flux export to the bottom water column,
and providing sufficient substrate for $CH_4$ production in the bottom layer (Gärdes et
al., 2011). Meanwhile, increased cell aggregation reduces respiration and $CO_2$
production in the upper water column (Hopkinson and Vallino, 2005). These could
explain why a higher positive interaction strength (number of links) between
phytoplankton and bacterioplankton was found in extreme carbon groups compared
with normal groups (Fig. 6). Hence, autochthonous OC critically influenced the fate
of extreme values of $CH_4$ and $CO_2$ through the interaction between PP and HB.
**5. Conclusion**
In the upper Yangtze's river-reservoir system, the normal $CH_4$ and $CO_2$
concentrations and fluxes were primarily contributed by the large input of
allochthonous OC. In contrast, the extreme values of $CH_4$ and $CO_2$ concentrations and
fluxes were mainly contributed by autochthonous OC. The picophytoplankton and
heterotrophic bacteria and their interactions were important ecological factors and
processes affecting extreme values of $CH_4$ and $CO_2$. Intensified interactions between
PP and HB, due to an increase in the trophic state, which strongly controlled the
generation and decomposition of autochthonous POC, concomitantly promoting the
extreme production and emissions of $CH_4$ and $CO_2$. Our findings provide a new



mechanism based on the interaction of picophytoplankton and heterotrophic bacteria,
which would contribute to a deeper understanding of extreme carbon emissions in the
river-reservoir ecosystem.
**Supplements**
Supplementary material associated with this article can be found on the
additional files.
**Data availability**
The raw sequencing data has been deposited to the National Center for
Biotechnology Information (NCBI) Sequence Read Archive (SRA) database under the
following BioProject accession numbers: PRJNA1188301 for 23S rRNA sequencing
data (http://www.ncbi.nlm.nih.gov/bioproject/1188301) and PRJNA1188367 for 16S
rRNA sequencing data (http://www.ncbi.nlm.nih.gov/bioproject/1188367).
**Author contributions**
ZL conceived this study and acquired the research funds. LL provided genome
sequence data. YX supervised the flow cytometry analysis. ZL and DW
conceptualized the study. FL, QT, YX and LL process data. FL and QT conducted
formal analysis. CL and XW assisted with the analysis. FL wrote the manuscript. ZL
revised the manuscript. All authors approved the manuscript.



**Competing interests**

The contact author has declared that none of the authors has any competing interests.

**Disclaimer**

Publisher's note: Copernicus Publications remains neutral with regard to jurisdictional claims made in the text, published maps, institutional affiliations, or any other geographical representation in this paper. While Copernicus Publications makes every effort to include appropriate place names, the final responsibility lies with the authors.

**Acknowledgements**

We acknowledge all of the partner projects listed in the financial support. We also thank Mr. Wei Tan, Ms. Xin Chen, and Mr. Qi Zhang who participated in sample collection and conducted laboratory analysis of water samples.

**Financial support**

This research was supported by the National Key Research and Development Program (2022YFC3203504). The National Natural Science Foundation of China (Project No. U2340222) also provided funding support.

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



**Figures**

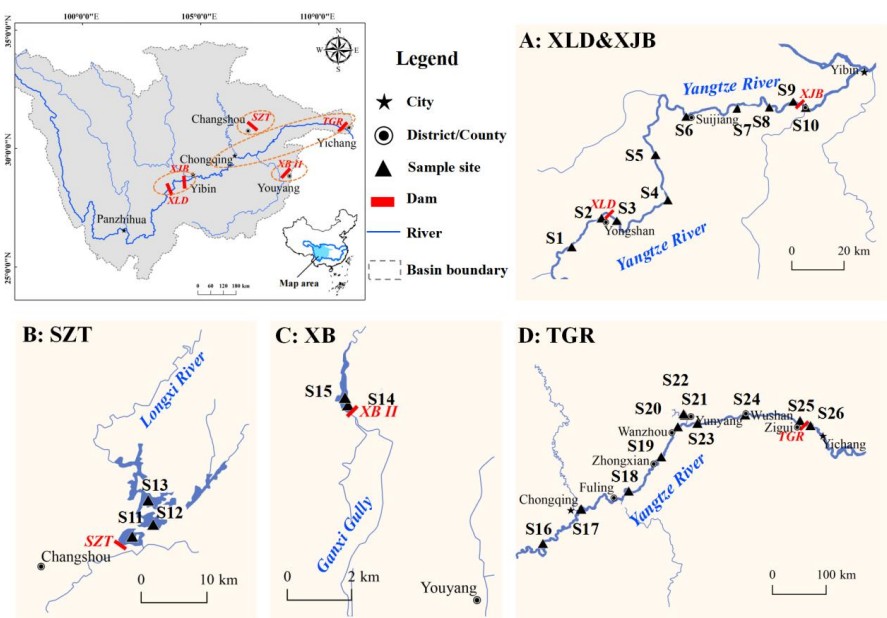


**Fig. 1.** Location of sampling sites in five reservoirs in the upper Yangtze River. Detailed
information on sampling sites in five reservoirs is shown in Table S2.

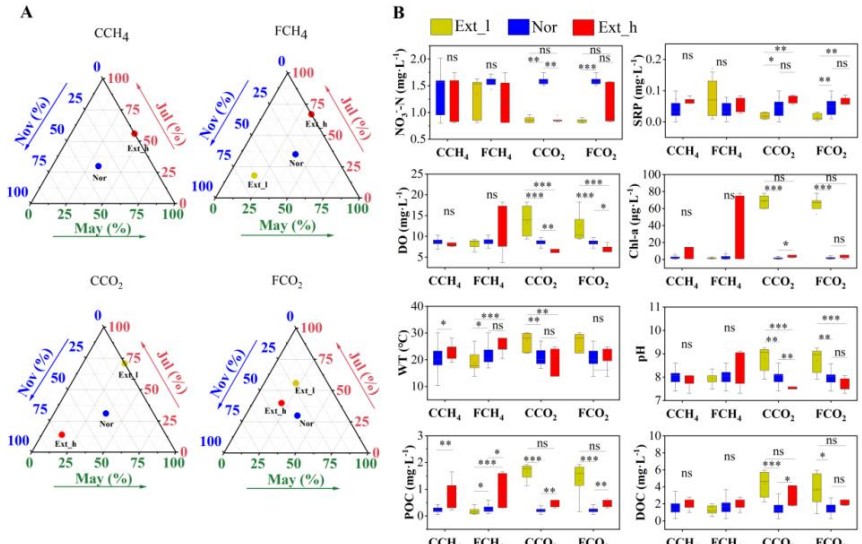

**Fig. 2.** Comparison of months and environmental parameters in extreme and normal groups. Panel A Ternary plots showing the percentage of months (May, July, and November) in which the extremely low (Ext_l), normal (Nor), and extremely high (Ext_h) values of $CH_4$ and $CO_2$ concentrations and fluxes occurred. The yellow, blue, and red dots represent the Ext_l, Nor, and Ext_h groups. Panel B Characteristics of $NO_3^-$-N, SRP, DO, Chl-a, WT, pH, POC, and DOC in surface water. The yellow, blue, and red boxes represent environmental parameters in the extremely low (Ext_l), normal (Nor), and extremely high (Ext_h) groups of $CH_4$ and $CO_2$ concentrations and fluxes, respectively. Asterisks indicate significant difference: * $p < 0.05$, ** $p < 0.01$, *** $p < 0.001$.



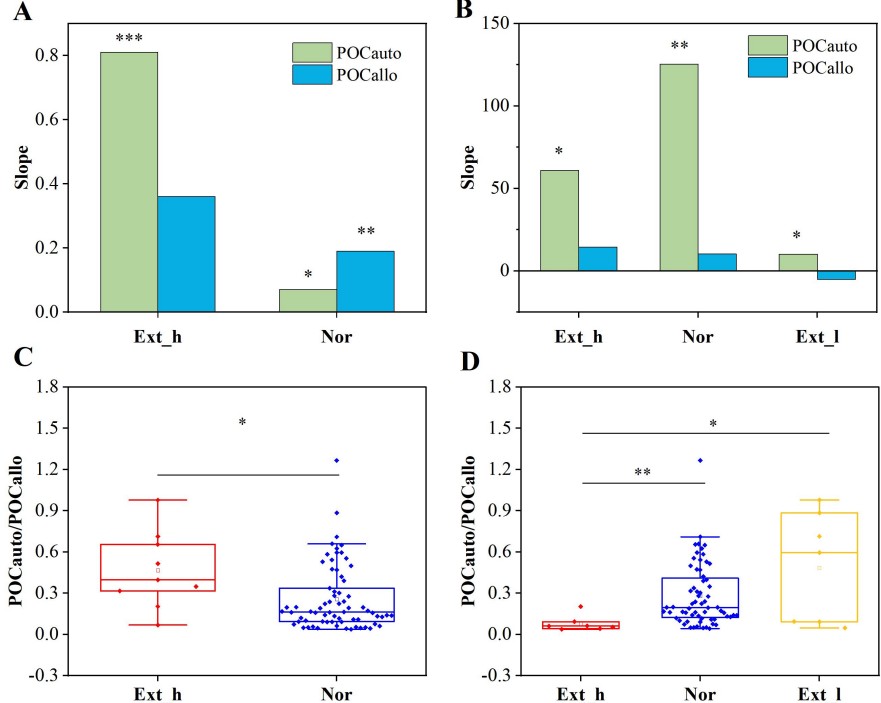

**Fig. 3.** Panels A and B Slope values of POC concentrations (autochthonous POC, allochthonous POC) linear regression analysis with carbon concentrations ($CCH_4$, $CCO_2$) in extremely high (Ext_h), normal (Nor) and extremely low (Ext_l) group, respectively. Panels C and D The concentration ratio of autochthonous POC to allochthonous POC (POCauto/POCallo) in extremely high (Ext_h), normal (Nor) and extremely low (Ext_l) groups of $CCH_4$ and $CCO_2$. Asterisks above the bar chart and box plot indicate significant levels: * $p < 0.05$, ** $p < 0.01$, *** $p < 0.001$.



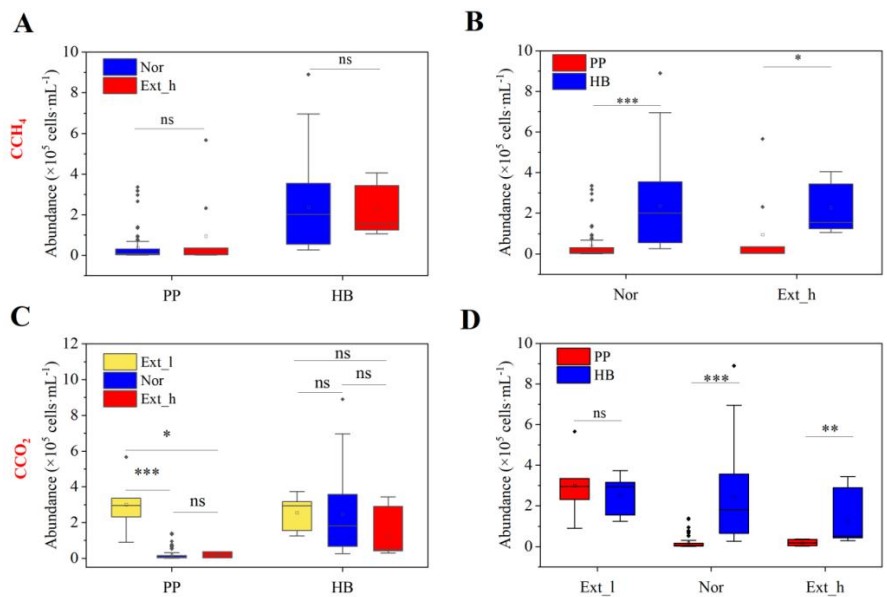

**Fig. 4.** Abundance of picophytoplankton and heterotrophic bacteria in the extremely high (Ext_h),

normal (Nor) and extremely low (Ext_l) groups of $CH_4$ (A, B) and $CO_2$ (C, D) concentrations.

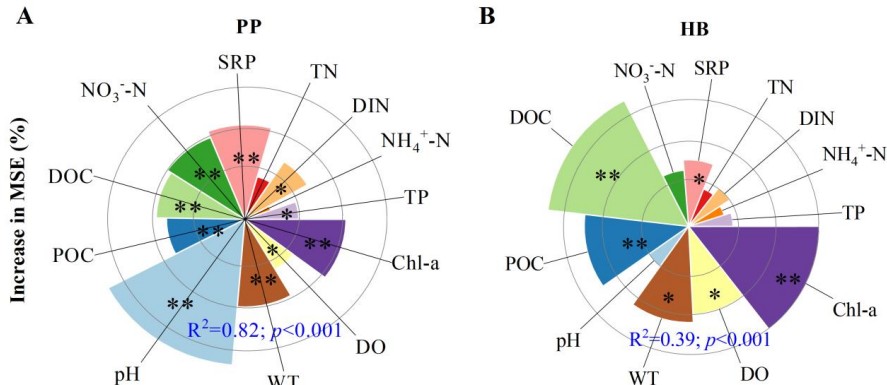

**Fig. 5.** Environmental predictors of picophytoplankton (A) and heterotrophic bacteria abundance

(B). Random forest modelling importance of environmental predictors on picophytoplankton and

heterotrophic bacteria abundance were estimated by percentage increases in the mean squared

error (%IncMSE). Significance levels of each predictor are as follow: * $p < 0.05$ and ** $p < 0.01$.

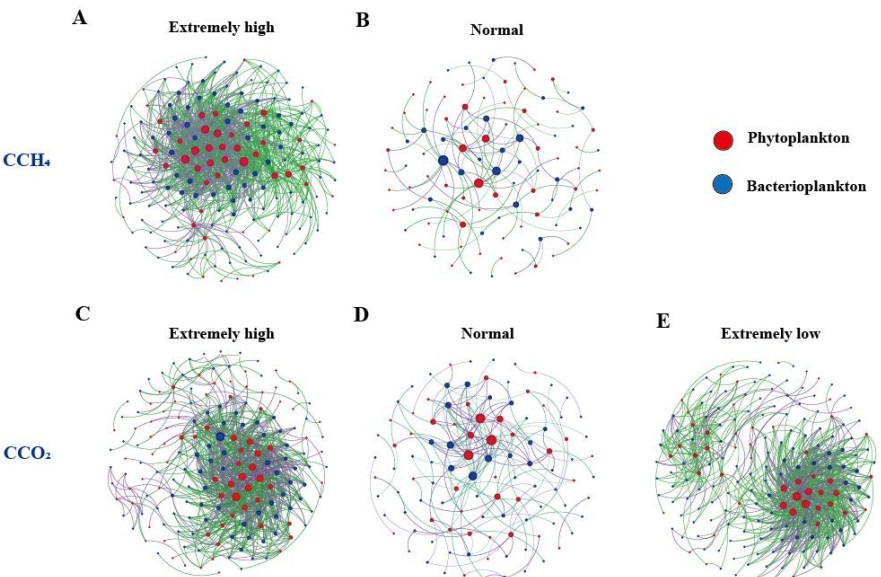

**Fig. 6.** Co-occurrence networks of phytoplankton and bacterioplankton communities based on correlation analysis. Panels A and B Co-occurrence patterns of phytoplankton-bacterioplankton interaction network in extremely high and normal groups of $CH_4$ concentration ($CCH_4$). Panels C, D, and E Co-occurrence patterns of phytoplankton-bacterioplankton interaction network in extremely high, normal, and extremely low groups of $CO_2$ concentration ($CCO_2$). Each line represents a significant correlation between the two taxa, with the green lines representing positive correlations and the violet lines representing negative correlations. The number of links represents the strength of interactions between phytoplankton and bacterioplankton. The red and blue nodes in each network represent phytoplankton and bacterioplankton, respectively. The size of each node is proportional to the number of connections (that is, degree).



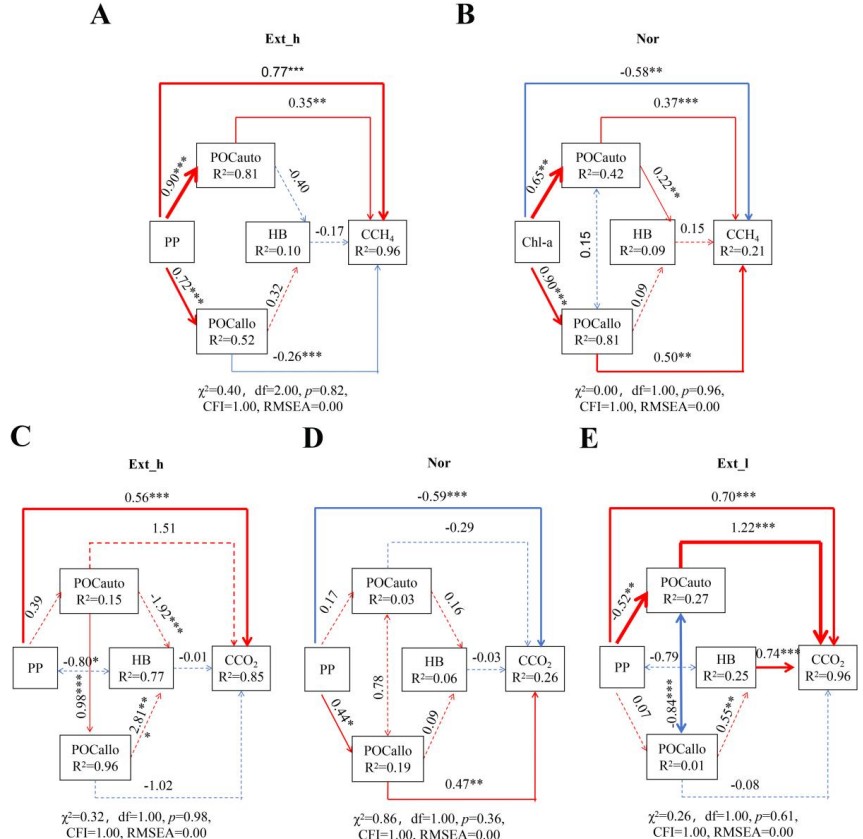

**Fig. 7.** Structural equation modeling (SEM) describing selected variables' effects on the concentration of $CH_4$ (A, B) and $CO_2$ (C, D, E) in the extremely high (Ext_h), normal (Nor) and extremely low (Ext_l) group, respectively. Numbers adjacent to arrows are standardized path coefficients and indicative of the effect size of the relationship. Solid arrows indicate significant paths (* $p < 0.05$, ** $p < 0.01$, *** $p < 0.001$), and dashed lines represent non-significant paths. The red and blue arrows indicate positive and negative path coefficients, respectively. The width of the arrows represents the strength of relationships. $R^2$ denotes the percentage of variance explained by the model. PP, picophytoplankton; HB, heterotrophic bacterial; POCauto, autochthonous POC; POCallo, allochthonous POC; $CCH_4$ and $CCO_2$ were the concentrations of $CH_4$ and $CO_2$.



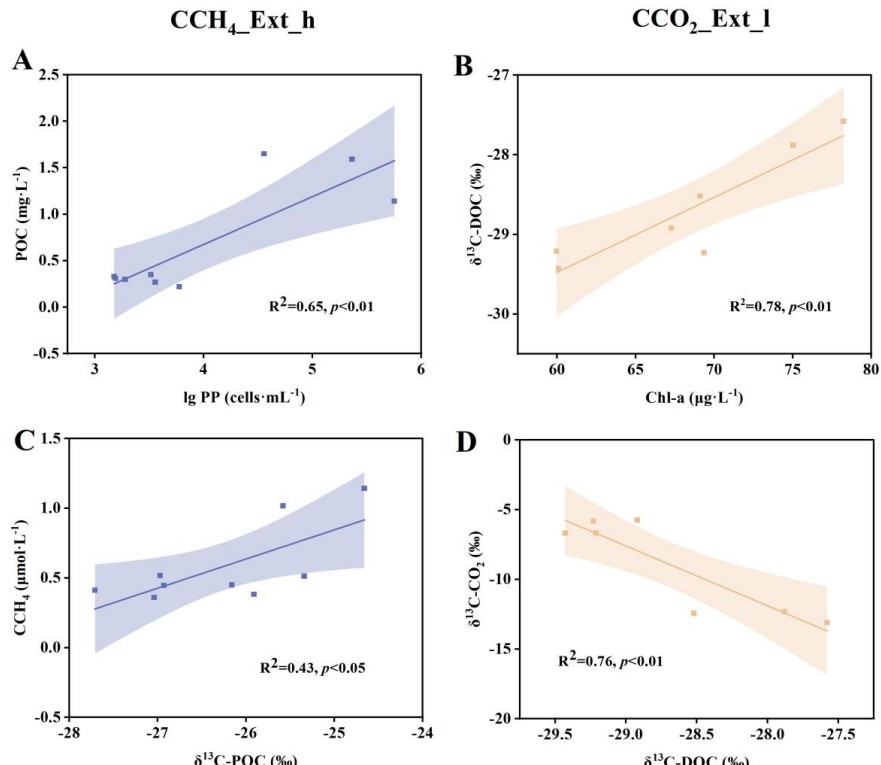

**Fig. 8.** Scatter plots of PP versus POC (A) and $\delta^{13}$C-POC versus $CCH_4$ (C) in the extremely high (Ext_h) group of $CCH_4$. Scatter plots of Chl-a versus $\delta^{13}$C-DOC (B) and $\delta^{13}$C-DOC versus $\delta^{13}$C-$CO_2$ (D) in the extremely low (Ext_l) group of $CCO_2$. The violet and yellow areas represent 95% confidence intervals.



**Tables**

**Table 1**

Concentrations and fluxes of $CO_2$ and $CH_4$ in the extremely low (Ext_l), normal (Nor) and extremely high (Ext_h) groups.

|  |  | Mean | SE | Range | N |
|---|---|---|---|---|---|
| Ext_l | $CCH_4$ | - | - | - | - |
|  | $FCH_4$ | 0.01 | 0.00 | 0.01-0.02 | 17 |
|  | $CCO_2$ | 12.62 | 1.49 | 8.46-18.01 | 7 |
|  | $FCO_2$ | 0.05 | 0.83 | -4.08-3.76 | 9 |
| Nor | $CCH_4$ | 0.03 | 0.00 | 0.00-0.12 | 69 |
|  | $FCH_4$ | 0.10 | 0.01 | 0.03-0.29 | 55 |
|  | $CCO_2$ | 44.25 | 1.33 | 24.28-64.63 | 64 |
|  | $FCO_2$ | 25.72 | 1.16 | 7.24-42.39 | 64 |
| Ext_h | $CCH_4$ | 0.19 | 0.02 | 0.13-0.33 | 9 |
|  | $FCH_4$ | 0.61 | 0.14 | 0.36-1.09 | 6 |
|  | $CCO_2$ | 78.66 | 2.91 | 71.92-92.74 | 7 |
|  | $FCO_2$ | 62.71 | 8.56 | 50.01-93.85 | 5 |

**Note:** $CCH_4$ and $CCO_2$ denote the concentrations of $CO_2$ and $CH_4$ in the surface water ($\mu mol \cdot L^{-1}$), respectively. $FCH_4$ and $FCO_2$ denote the fluxes of $CH_4$ and $CO_2$ across the water-air interface ($mmol \cdot m^{-2} \cdot d^{-1}$), respectively. -: not exist; SE: standard error; N: the number of observations.



**Table 2**

TLI values in extreme and normal groups of $CH_4$ and $CO_2$ concentrations and fluxes

|  | Groups | Mean | SE | Range | Trophic state |
|---|---|---|---|---|---|
|  | $CCH_4\_Ext\_l$ | - | - | - | - |
|  | $CCH_4\_Nor$ | 44.99 | 1.07 | 33.91-70.03 | M |
|  | $CCH_4\_Ext\_h$ | 50.54 | 4.19 | 41.13-70.81 | E |
|  |  |  |  |  |  |
|  | $FCH_4\_Ext\_l$ | 42.04 | 0.95 | 36.96-49.14 | M |
|  | $FCH_4\_Nor$ | 45.79 | 1.29 | 33.91-70.03 | M |
|  | $FCH4\_Ext\_h$ | 54.32 | 5.79 | 41.13-70.81 | E |
| TLI |  |  |  |  |  |
|  | $CCO_2\_Ext\_l$ | 69.11 | 0.50 | 67.41-70.80 | E |
|  | $CCO_2\_Nor$ | 42.39 | 0.69 | 33.91-67.87 | M |
|  | $CCO_2\_Ext\_h$ | 51.71 | 1.80 | 47.13-58.81 | E |
|  |  |  |  |  |  |
|  | $FCO_2\_Ext\_l$ | 62.66 | 4.29 | 39.49-70.81 | E |
|  | $FCO_2\_Nor$ | 42.84 | 0.71 | 33.91-67.87 | M |
|  | $FCO_2\_Ext\_h$ | 50.65 | 3.70 | 38.93-58.81 | E |

**Note:** -: not exist.



**Table 3**

Topological properties of co-occurrence network of phytoplankton-bacterioplankton interaction in the extremely low (Ext_l), normal (Nor), and extremely high (Ext_h) groups of CH$_4$ (CCH$_4$) and CO$_2$ concentrations (CCO$_2$).

| Network metrics | CCH$_4$ | | CCO$_2$ | | |
|---|---|---|---|---|---|
| | Nor | Ext_h | Ext_l | Nor | Ext_h |
| Number of nodes | 101 | 173 | 184 | 108 | 175 |
| Number of edges (Interaction strength) | 104 | 977 | 800 | 171 | 841 |
| Number of positive edges | 59 (56.73%) | 630 (64.48%) | 543 (67.87%) | 76 (44.44%) | 561 (66.71%) |
| Number of negative edges | 45 (43.27%) | 347 (35.52%) | 257 (32.12%) | 95 (55.56%) | 280 (33.29%) |
| Modularity | 0.752 | 0.292 | 0.404 | 0.558 | 0.341 |
| Average degree | 2.059 | 11.295 | 8.696 | 3.167 | 9.611 |



**Table 4**

Results of multiple linear regression analysis relating GHGs with POC from different sources in

the $CCH_4$_ Ext_h and $CCO_2$_Ext_l group, respectively.

| Group | Equations | $R^2$ | $p$ | Significance level T-test | |
|---|---|---|---|---|---|
| | | | | $t_1$ | $t_2$ |
| Ext_h | $CH_4=1.11 (POCauto)-0.10 (POCallo)+0.38$ | 0.74 | ** | 4.34** | -0.61 |
| Ext_l | $LgCO_2=1.36[Lg(POCauto+1)]+0.29[Lg(POCallo +1)]+0.78$ | 0.70 | * | 3.02* | 0.55 |

**Note:** Asterisks indicate significant levels: * $p < 0.05$, ** $p < 0.01$, *** $p < 0.001$.