# Peer review of "2 3 Title: Extreme carbon fluxes may result from autochthonous particulate organic carbon regulated by the interactions between picophytoplankton and heterotrophic 4 5 bacteria in river-reservoir systems 6 Authors: Fang Luo 1,2,3, Zhe Li 2,3, Qiong Tang 1,2,3, Yan Xiao 2,3, Lunhui Lu 2,3, 7 Dianch"

_EGUsphere, 2025_

## Author Comment (AC1)

**Response to Reviews for the manuscript #egusphere-2025-156**

We sincerely thank Referee #1 and the editor for reviewing and evaluating our manuscript entitled "Extreme carbon fluxes may result from autochthonous particulate organic carbon regulated by the interactions between picophytoplankton and heterotrophic bacteria in river-reservoir systems" (ID: egusphere-2025-156), and for providing valuable and constructive comments. These comments help us to improve the quality of the manuscript. We have revised the manuscript according to Referee #1's comments and provided detailed point-by-point responses to the comments. In the following, Referee #1's comments are shown in blue, and our responses are presented in black. The corresponding revisions have been highlighted in yellow in the revised manuscript.

The Referee #1's general and specific comments and our responses are as follows:

General comments included 3 points.

1) Lack of study context : The Abstract lacks context to what is studied. Extreme fluxes are not described until Line 67 — well into the introduction. One could assume this means a one time event rather than over the course of an algal bloom. Adding a brief description would provide valuable context for the reader. Similarly, it is awkward to say river-reservoir without providing this detail earlier in the abstract. It would be useful to specifically state earlier what your study system is.

We thank you for pointing this out. We have provided study context about extreme fluxes in the *Abstract* in Lines 23-25 of the clean version of the revised

manuscript. We have also added that the system we studied was the river-reservoir system in the section *Abstract* in Lines 29-31 of the clean version of the revised manuscript. Specific revisions are below.

| **Revised manuscript** |
|---|
| Reservoir ecosystems are significant natural sources of atmospheric $CH_4$ and $CO_2$, while also receiving large amounts of nutrients and particulate organic carbon (POC) from various sources. Ecosystem-level events, particularly algal blooms due to eutrophication, significantly contribute to the occurrence of extremes in $CH_4$ and $CO_2$ concentrations and fluxes, likely linked to ecological factors and processes. The trophic interaction between picophytoplankton (PP) and heterotrophic bacteria (HB) plays a vital role in the carbon cycle within aquatic systems. However, it remains unclear which source of POC primarily drives carbon extremes in reservoirs and how PP-HB interactions influence extreme carbon emissions. Here, we investigated contributions of POC from different sources to extreme carbon emissions, along with the interaction between PP and HB in river-reservoir ecosystems. The evidence from isotope analysis further proved that the extreme carbon fluxes were strongly influenced by autochthonous POC rather than allochthonous POC. Network analysis showed that the positive interaction strength between phytoplankton and bacterioplankton in extreme carbon groups was higher than in normal carbon groups. The results of the structure equation modeling analysis also highlighted that the PP-HB interaction strongly drove the extreme carbon values. This study first introduced the probability statistics method to |

identify and classify high or low extreme carbon values. These findings also highlight the importance of PP and HB in carbon extreme emissions, and we hope our study can provide an important implication for integrating PP-HB interaction into predicting extreme carbon emissions in river-reservoir ecosystems.

2) Over-speculation and lack of data comparison in discussion: The $CO_2$ and $CH_4$ concentration and flux data are not presented in light of previous studies. As such, the reader has no context for these values unless they are familiar with specific studies themselves. There are loads of studies on $CO_2$ and $CH_4$ fluxes, concentrations, etc and the authors could shorten the current Discussion which is overly speculative in some areas (particularly the amplicon data which is only DNA, not related to function or activity, etc) to make room to discuss the actual values here and how they compare to other studies. There is no need to write so much about r- vs. K-strategists or correlation networks when the data only support correlations and not in situ activity that can be directly linked to fluxes or concentrations.

We sincerely appreciate the reviewer's constructive suggestions. $CH_4$ and $CO_2$ concentrations and fluxes in warm and cold seasons across reservoirs with different trophic states from this and previous studies were presented in Table 5. The $CO_2$ and $CH_4$ concentration and flux data comparisons with other studies have been added in the third paragraph of section 4.3 *Extreme and normal patterns of carbon emissions driven by picophytoplankton and heterotrophic bacteria*. The discussion of actual values in our study has been added in the last two paragraphs of section 4.3. We have also shortened the speculative discussion on the results related to r- vs. K-strategists

and correlation networks in section 4.3. Hence, we revised section 4.3 in Lines 577-648 of the clean version of the revised manuscript according to these suggestions. Specific revisions are below.

*4.3. Extreme and normal patterns of carbon emissions driven by picophytoplankton and heterotrophic bacteria*

Although some extreme $CO_2$ and $CH_4$ air-water fluxes could be induced by short-term physical processes, the short-term physical processes were not the focus of our study. We found that water temperature showed no difference between normal and extreme groups of $CO_2$ fluxes (Fig. 2B). Yet, the picophytoplankton abundance in the extremely low group of $CO_2$ concentration was significantly higher than that in the normal group of $CO_2$ concentration (Fig. 4C). These results suggested that the extreme and normal values of $CH_4$ and $CO_2$ were probably influenced by ecosystem response (such as microbial community composition or abundance variation), rather than physical factors (such as temperature and wind).

The dramatic fluctuations in the abundance of PP and HB are important ecological factors driving extreme values of carbon concentrations and fluxes. This can be primarily attributed to two key characteristics of PP and HB. First, their rapid response to environmental changes. When exposed to elevated temperatures and increased nutrient inputs, PP and HB respond quickly, often exhibiting explosive growth (Flombaum et al., 2013; Azam and Malfatti et al., 2007). In contrast, larger phytoplankton respond more slowly, and their abundance tends to remain relatively stable under changing environmental conditions (Litchman et al., 2007). Second, their strong capacity to decompose easily degradable OC, including autochthonous OC and labile fractions of allochthonous inputs. Based on their

growth rate and carbon utilization efficiency, microbial communities can be classified into two ecological functional groups, r- and K-selected species (Li et al., 2021). As r-strategists, picophytoplankton and heterotrophic bacteria exhibit a fast growth rate and a rapid response to labile C, whereas K-selected species are slow-growing, decompose recalcitrant OC more efficiently, and respond slowly to OC inputs (Li et al., 2021). This may explain the positive associations between picophytoplankton abundance and extreme $CH_4$ and $CO_2$ concentrations (Fig. 7). Hence, picophytoplankton and heterotrophic bacteria play essential roles in maintaining short-term extreme carbon emissions.

Seasonal comparisons of $CH_4$ and $CO_2$ concentrations and fluxes across reservoirs (Table 5) revealed that in eutrophic reservoirs, $CH_4$ concentrations and fluxes in the warm season were higher than those in the cold season, whereas $CO_2$ exhibited the opposite trend. These results were consistent with findings reported for lakes (Zhang et al., 2022; Sun et al., 2021; Wang et al., 2022), where frequent algal blooms significantly increase $CH_4$ emissions and enhance $CO_2$ uptake from the atmosphere in warm seasons. Together, these observations suggest that dramatic fluctuations in PP and HB abundance, induced by high nutrient levels and elevated water temperatures, may have contributed to extremely high $CH_4$ values and extremely low $CO_2$ values in eutrophic reservoirs.

Extremely high $CH_4$ concentrations and fluxes mainly occurred in July (Fig. 2A), and extremely high values of $CH_4$ concentration were positively influenced by PP abundance ($p<0.001$; Fig. 7A). During the algal blooming period, high nutrient

inputs and elevated temperatures facilitated the explosive growth of PP and HB in the warm season (Tang et al., 2017). The increased cell density of PP (Fig. S7) likely enhanced the possibility of interactions between PP and HB (Christie-Oleza et al., 2017). Increased PP-HB interactions facilitate cell aggregation, which increases carbon flux export to the bottom water column, thereby providing sufficient substrate for $CH_4$ production in the bottom layer (Gärdes et al., 2011). Additionally, eutrophication promotes positive interactions (i.e. cooperation) between PP and HB (Fig. S11), strengthening positive feedback between PP and HB (Coyte et al., 2015). These findings help explain why strong positive interaction strength (number of links) between phytoplankton and bacterioplankton was found in extreme carbon groups (eutrophic state) compared with normal groups (mesotrophic state) (Fig. 6). In eutrophic reservoirs, the efficient production and rapid decomposition of easily degradable autochthonous POC by PP and HB, respectively, accelerate $CH_4$ production in the short term (West et al., 2012; Gärdes et al., 2011), resulting in extremely high $CH_4$ concentrations and fluxes in the warm season (Fig. 2A). Indeed, a significant positive correlation between network degree (interaction strength) and $CH_4$ concentrations and fluxes was found under eutrophic conditions (Fig. S12). Such a positive correlation may support our third hypothesis; however, further validation through laboratory incubation experiments and functional gene analysis is still required.

The abundance of PP and HB had a significantly positive impact on extremely low $CO_2$ concentration ($p<0.001$; Fig. 7E), and extremely low $CO_2$ concentrations

and fluxes were predominantly observed during warm seasons (Fig. 2A). Increased PP-HB interactions enhance cell aggregation of PP, which increases downward carbon fluxes and thereby reduces $CO_2$ production in the upper water column (Hopkinson and Vallino, 2005). Moreover, the impact of algal blooms on $CO_2$ dynamics is co-determined by both $CO_2$ production (including external DIC input and the decomposition of OC) and consumption via primary production (Zhang et al., 2022). In winter, lower water temperature would inhibit PP reproduction, weaken photosynthesis of PP, and thus reduce $CO_2$ uptake (Ren et al., 2022). In contrast, during warm seasons, the explosive growth of PP in eutrophic reservoirs greatly enhances $CO_2$ uptake, and massive $CO_2$ consumption by primary production can offset or even exceed $CO_2$ production from external DIC input and microbial decomposition of autochthonous OC by HB (Zhang et al., 2022). Additionally, high $CO_2$ uptake is supported by high DO, Chl-a, and water temperature (Yang et al., 2021), consistent with the higher DO, Chl-a, and water temperature observed in the extremely low $CO_2$ groups (Fig. 2B). This helps explain why the eutrophic Shizitan Reservoir acted as a $CO_2$ sink in the warm season, but as a source of atmospheric $CO_2$ in the cold season (Table 5). Hence, autochthonous OC critically influenced the fate of extreme carbon values through the interaction between PP and HB.

**Table 5**

Concentrations and air-water fluxes of $CH_4$ and $CO_2$ in the investigated reservoirs compared with those in other reservoirs of different trophic states.

| Reservoir name | Trophic state | Climate zone | Chl-a (Warm/Cold) | CCH4 (Warm/Cold) | FCH4 (Warm/Cold) | CCO2 (Warm/Cold) | FCO2 (Warm/Cold) | Reference |
|---|---|---|---|---|---|---|---|---|
| Reservoir Xiaoba II | Oligotrophic | Subtropical | 2.90±0.08/ 1.76±0.22 | 0.12±0.01/ 0.20±0.05 | 0.11±0.01/ 0.15±0.07 | 29.68±0.13/ 38.33±0.20 | 14.90±0.62/ 1.96±1.27 | This study |
| Reservoir Chapéu d'Uvas | Oligotrophic | Tropical | 11.5[a] | - | 2.50/1.60 | - | 11.00/1.60 | Linkhorst et al., 2021; Paranaíba et al., 2021 |
| Reservoir Panjiakou | Mesotrophic | Temperate | 3.96/0.21 | 0.152/0.146 | 0.01/0.03 | 187.52/195.07 | 17.91/32.00 | Yang et al., 2021 |
| Reservoir Daheiting | Mesotrophic | Temperate | - | 0.19±0.12/ 0.41±0.26 | 0.07±0.05/ 0.12±0.08 | 72.75±67.49/ 394.64±104.13 | 19.45±18.98/ 115.75±30.00 | Gong et al., 2019 |
| Reservoir Xiluodu | Mesotrophic | Subtropical | 2.75±0.92/ 1.03±0.12 | 0.04±0.00/ 0.02±0.00 | 0.03±0.00/ 0.01±0.00 | 43.14±3.34/ 37.85±0.43 | 21.68±5.25/ 17.61±2.78 | This study |
| Reservoir Qianxiahu | Mesotrophic | Subtropical | 5.26±3.12/ 4.68±2.54 | 0.25±0.10/ 0.11±0.07 | 0.17±0.09/ 0.02±0.04 | 3.05±1.51/ 6.54±1.94 | -47.4±29.4/ -8.69 ± 8.30 | Zhang et al., 2024 |
| Reservoir Shizitan | Eutrophic | Subtropical | 65.45±4.83/ 2.67±0.55 | 0.07±0.03/ 0.02±0.01 | 0.08±0.03/ 0.01±0.00 | 9.18±0.63/ 80.04±4.36 | -2.68±1.21/ 42.07±7.74 | This study |
| Reservoir Andean | Eutrophic | Subtropical | - | 1.44±0.41/ 0.50±0.39 | 2.19±0.47/ 0.75±0.04 | 148.4±50.57/ 500.57±41.26 | -6.54±50.72/ -18.48±5.34 | Eliana et al., 2024 |
| Reservoir Yuqiao | Eutrophic | Temperate | 46.82±19.17[a] | 0.16±0.02[a] | 1.44/0.12 | - | - | Zhong et al., 2023 |
| Reservoir Funil | Eutrophic | Tropical | 29.5[b] | - | 0.06±0.00 | - | 0.04±0.00 | Paranaíba et al., 2021 |

**Note:** a The average value over study period. CCH4 and CCO2 denote the concentrations of $CO_2$ and $CH_4$ in the surface water ($\mu mol \cdot L^{-1}$), respectively. FCH4 and FCO2 denote the fluxes of $CH_4$ and $CO_2$ across the water-air interface ($mmol \cdot m^{-2} \cdot d^{-1}$), respectively. Warm denotes the warm season, i.e. spring, summer, or autumn; Cold denotes winter. Chl-a denotes the concentrations of chlorophyll a ($\mu g \cdot L^{-1}$). -: not exist.

3) Inconsistent terminology: Sometimes the paper uses acronyms for samples, other times it does not. This adds confusion to the writing. I would choose one and stick with that style throughout.

We thank you for pointing this out. To maintain consistency throughout the text, all sample names in the article are not abbreviated. All acronyms for sample names in the text, such as Ext_h, Ext_l, and Nor, have been deleted from the revised manuscript. We have replaced the "CCH$_4$_Ext_h" with "extremely high group for CH$_4$" and "CCO$_2$_Ext_l" with "extremely low group for CO$_2$" in Lines 525-526 of the clean version of the revised manuscript. Specific revisions are below.

| Revised manuscript |
| --- |
| In this study, we found that the abundances of both picophytoplankton and heterotrophic bacteria were slightly higher in the ==extremely high group of CH$_4$ concentration and the extremely low group of CO$_2$ concentration (eutrophic state) than those in the corresponding normal groups== (mesotrophic state) (Fig. 4; Table 2; Fig. S7). |

Specific comments included 19 points.

1) Line 80: Providing significant.

We thank you for pointing this out. We added the significance of picophytoplankton in the carbon cycle in Lines 89-91 of the clean version of the revised manuscript. Specific revisions are below.

| Revised manuscript |
| --- |
| The minor component of the planktonic communities (Stockner and Antiam, |

1986), the picoplankton (defined by a cell size of 0.2-2 μm) (Sieburth et al., 1978), mainly includes autotrophic picophytoplankton and heterotrophic bacteria (Stockner, 1988). Picophytoplankton are active and critical primary producers in aquatic ecosystems due to their wide distribution, rapid growth rates, and metabolic capabilities (Stockner, 1988). Picophytoplankton play a crucial role in primary production and carbon fixation in global aquatic ecosystems (Platt et al., 1983; Stockner, 1988).

2) Line 81: This is an awkward statement that could be re-phrased. Also is there any specific quantitative information you can provide other than "more CO2"? Related —the next statement is about picophytoplankton generally and not blooms specifically so you might consider reorganizing here for clarity.

We thank you for pointing this out. We have deleted "during an algal bloom" and provided the specific contribution proportion of picophytoplankton to $CO_2$ fixation in oligotrophic oceans. The description can be found in Lines 93-96 of the clean version of the revised manuscript. Specific revisions are below.

| Revised manuscript |
| --- |
| In the ocean, picophytoplankton can contribute 50-90% of primary productivity (Poulton et al., 2006), significantly providing autochthonous organic carbon to heterotrophic bacteria. In addition, picophytoplankton contribute substantially to $CO_2$ fixation in aquatic ecosystems. Especially in oligotrophic oceans, $CO_2$ fixation by picophytoplankton, such as *Prochlorococcus and Synechococcus*, can account for up to 60% of the total $CO_2$ fixation (Flombaum et |

al., 2013). This is because picophytoplankton have higher growth rates and are more effective in nutrient and light acquisition than larger phytoplankton (Irion et al., 2021).

3) Line 86: Why is tiny here? Are these smaller than "defined by a cell size of 0.2-2 μm"?

We thank you for pointing this out. We understand that the use of "tiny" is ambiguous here, therefore we have removed "tiny" in the revised manuscript.

4) Line 89: In water? Or in all environments?

We thank you for pointing this out. The research findings here are focused on the marine ecosystem. The phrase "In the ocean" has been added in Line 103 of the clean version of the revised manuscript. Specific revisions are below.

| Revised manuscript |
| --- |
| On the other hand, heterotrophic bacteria decompose organic carbon, transferring different sources of OC into $CH_4$ or $CO_2$ (Guillemette et al., 2013). It was reported that heterotrophic bacteria can consume 20-60% of the total primary production (Williams, 1981) in the ocean. |

5) Line 108: autotrophic

We thank you for pointing this out. We have corrected "autrophic" to "autotrophic" in Line 108 of the clean version of the revised manuscript.

6) Line 115: Why the quotes for "physical interactions"? Do you need to define this? Does it mean something other than what is stated?

We thank you for pointing this out. In our study, we indeed don't need to define

"physical interactions". We have corrected it in Line 129 of the clean version of the revised manuscript. Specific revisions are below.

| Revised manuscript |
|---|
| First, the cooperative relationship between picophytoplankton and heterotrophic bacteria produces strong coupling and positive feedback between these two organisms, increasing microbial metabolic efficiency and full utilization of OC (Coyte et al., 2015). Second, the interactions between picophytoplankton and heterotrophic bacteria mediate the level of aggregation of picophytoplankton biomass, which manipulates downward C flux (Seymour et al., 2017). |

7) Line 128: Why the quotes again?

We thank you for pointing this out. We understand the confusion and have replaced "specific participants" with "PP and HB" in Line 141 of the clean version of the revised manuscript. Specific revisions are below.

| Revised manuscript |
|---|
| Little is known about the dynamics of $CH_4$ and $CO_2$ production and emissions driven by the interactions between PP and HB in freshwaters. |

8) Line 137: Reference needed?

We thank you for pointing this out. We have added relevant references regarding the research on excessive carbon emissions from reservoirs in Line 151 of the clean version of the revised manuscript. Specific revisions are below.

| Revised manuscript |
|---|
| Over the past two decades, there has been increasing concern about the |

excessive carbon emissions from reservoirs (Beaulieu et al., 2019; Wei et al., 2025).

9) Line 139, Line 142, Line 154: As I mentioned above, a definition of extremes is needed.

We thank you for pointing this out. We have added a definition of extreme values in the second paragraph of the *Introduction* in Lines 68-75 of the clean version of the revised manuscript. Specific revisions are below.

| **Revised manuscript** |
| --- |
| The term "extreme values" is commonly used in meteorology and hydrology (Sun and Qin, 1989; Ding and Jiang, 2009). Extreme values refer to values that fall within the 5% or 10% range at the tails of both ends of the probability distribution curve—that is, values corresponding to the low frequency, such as extremely high or low wind speeds and air temperatures (Ding and Jiang, 2009). Although extremes of $CH_4$ and $CO_2$ concentrations or fluxes are not frequently detected, their extreme values may reflect ecosystem-level states and processes. The air-water $CH_4$ flux was expected to exhibit extremely high values during algal blooms, concurrently with low levels of surface water $CO_2$ concentrations, leading to an apparent $CO_2$ sink during the blooming period (Sun et al., 2021). |

10) Line 142: Can you provide specific numbers for what is high or low?

We thank you for pointing this out. We employed the Pearson Type III probability distribution to determine the extreme and normal values of $CH_4$ and $CO_2$ concentrations and fluxes. Values corresponding to the 10% probability range at both tails of the theoretical curve were selected as extreme values, while those within the

10–90% probability range were defined as normal values. The empirical frequencies of the samples were calculated based on $CH_4$ and $CO_2$ concentrations and fluxes. Meanwhile, the sample data points and the theoretical Pearson Type III curve representing the population were plotted on Hessian probability grid paper (Fig. S1).

For $CH_4$ concentrations, the threshold values for the extremely high and extremely low $CH_4$ concentrations were set at 0.131 and 0.004, respectively. The number of samples corresponding to the extremely high, normal, and extremely low groups of $CH_4$ concentrations was 9, 69, and 0, respectively. For $CH_4$ fluxes, the threshold values for the extremely high and extremely low $CH_4$ fluxes were set at 0.309 and 0.023. The numbers of samples in the extremely high, normal, and extremely low $CH_4$ flux groups were 6, 55, and 17, respectively.

For $CO_2$ concentrations, the threshold values for extremely high and extremely low $CO_2$ concentrations were set at 66.985 and 22.868, respectively. The numbers of samples in the extremely high, normal, and extremely low $CO_2$ concentration groups were 7, 64, and 7, respectively. For $CO_2$ flues, the threshold values for extremely high and extremely low $CO_2$ fluxes were set at 46.586 and 7.109, respectively. The corresponding number of samples was 5, 64, and 9, respectively.

The supplementary material Text S3 *Determination of extreme and normal values of greenhouse gases* provides the threshold values and sample numbers for the above three groups of $CH_4$ and $CO_2$ concentrations and fluxes.

11) Line 148: The hypotheses — is this about concentration and flux?

We thank you for pointing this out. We have added "fluxes" in Lines 164-168 of

the clean version of the revised manuscript. Specific revisions are below.

| Revised manuscript |
|---|
| Therefore, we hypothesized that (1) autochthonous organic carbon (OC) in river-reservoir systems greatly contributes to the occurrences of extreme values of $CH_4$ and $CO_2$ concentrations ==and fluxes==; (2) terrigenous OC contributes to the normal values of $CH_4$ and $CO_2$ concentrations ==and fluxes==; and (3) The interaction of autotrophic picophytoplankton (PP) and heterotrophic bacteria (HB) could be intensified with an increase in trophic state, thus promoting the production of extreme values of $CH_4$ and $CO_2$ ==concentrations and fluxes==. |

12) Line 154: these hypotheses

We thank you for pointing this out. We have corrected "the hypothesis" to "these hypotheses" in Line 169 of the clean version of the revised manuscript.

13) Line 216, Line 243: Acidification has been show to impact 15N signals.

We thank you for pointing this out. After acidification, we repeatedly rinsed membranes containing particulate matter with deionized water until the pH reached 7, and then dried them prior to stable isotope analysis. The description of HCl removal and particulate matter drying after acidification has been added to the last paragraph of section 2.2 *Physicochemical parameters* in Lines 227-233 of the clean version of the revised manuscript. Specific revisions are below.

| Revised manuscript |
|---|
| ==The membranes containing particulate matter were dried at 65 ℃ for 48 h. Subsequently, the dried membranes were fumigated with HCl (12 M) for 12 h to== |

remove particulate inorganic carbon, rinsed with deionized water until the pH reached 7, and then dried again at 65 ℃ (Xie et al., 2020) prior to stable isotope analysis. The concentrations and stable isotope values of POC and PON were measured using an elemental analyzer coupled with a stable isotope mass spectrometer (Thermo Fisher Scientific® Flash HT-Delta V Advantage, USA).

14) Line 230: Do you mean from duplicate samples or what was duplicated here?

We thank you for pointing this out. Here, the purpose of this sentence was to describe the DNA extraction in duplicate, but the expression was incorrect. We understand that it may cause confusion, and we have made the correction in the second paragraph of section 2.3 *Analysis of microbial communities* in Lines 246-247 of the clean version of the revised manuscript. Specific revisions are below.

| **Revised manuscript** |
| --- |
| Genomic DNA was extracted in duplicate from the filters using the FastDNA SPIN kit (Mo Bio laboratories®, USA), following the manufacturer's instructions. The duplicate DNA extracts were mixed for the following PCR amplification. |

15) Line 329: Doesn't the previous statement imply the opposite — low, July, high, November. This is very confusing as presented.

We felt sorry for the confusion that was caused. We have made the correction in Lines 343-346 of the clean version of the revised manuscript. Specific revisions are below.

| **Revised manuscript** |
| --- |
| These results exhibited that extremely high $CH_4$ and extremely low $CO_2$ |

values mostly appeared in July, supporting the inference that cell aggregation mediated by the PP-HB interactions drives these extremes, especially during the summer algal bloom period.

16) Line 277: Why introduce Ext_h, Nor here but not use it before?

We thank you for pointing this out. To maintain consistency throughout the text, all sample names in the article are not abbreviated. All acronyms for sample names in the text, such as Ext_h, Ext_l, and Nor, have been deleted from the revised manuscript. We have replaced the "CCH$_4$_Ext_h" with "extremely high group for CH$_4$" and "CCO$_2$_Ext_l" with "extremely low group for CO$_2$" in Lines 525-526 of the clean version of the revised manuscript. Specific revisions are below.

| **Revised manuscript** |
| --- |
| In this study, we found that the abundances of both picophytoplankton and heterotrophic bacteria were slightly higher in the extremely high group of CH$_4$ concentration and the extremely low group of CO$_2$ concentration (eutrophic state) than those in the corresponding normal groups (mesotrophic state) (Fig. 4; Table 2; Fig. S7). |

17) Line 391: What do you mean by certain influence?

We thank you for pointing this out. The sentence is intended to describe the influence of some environmental factors, such as DOC, POC, and WT, on the abundance of HB to some extent, but the expression was incorrect. We understand that it may cause confusion, and we have made the correction in the last paragraph of section 3.3 *Picophytoplankton and heterotrophic bacteria abundance across extreme*

*and normal groups* in Lines 403-405 of the clean version of the revised manuscript. Specific revisions are below.

| Revised manuscript |
|---|
| The main environmental predictors influencing PP abundance were TP, SRP, DIN, $NO_3^-$-N, Chl-a, DO, WT, pH, POC, and DOC ($p < 0.05$; Fig. 5A). The main environmental variables, such as DOC, POC, WT, DO, Chl-a, and SRP, were found to significantly influence HB abundance ($p < 0.05$; Fig. 5B). The variance explanation of environmental factors for the abundance of PP ($R^2$=0.82) was higher than that of HB ($R^2$=0.39). |

18) Line 484: accounts for a large proportion of the POC

We thank you for pointing this out. We have made the correction in the third paragraph of section 4.1 *Contributions of POC from different sources to CH₄ and CO₂* in Lines 498-499 of the clean version of the revised manuscript. Specific revisions are below.

| Revised manuscript |
|---|
| Nevertheless, because allochthonous POC accounts for a large proportion of POC in aquatic ecosystems, it may support long-term $CH_4$ accumulation and emissions (Berberich et al., 2020). |

19) Line 580: Correlation is not the same thing as causation. Especially from 16S rRNA (DNA) data. These statements should reflect that this is only a correlation in the network analyses.

We sincerely appreciate the reviewer's insightful comments. Such a positive correlation may support the third hypothesis that increased interactions between

picophytoplankton and heterotrophic bacteria promote the extreme values of carbon.

However, the correlation results only indicate that PP-HB interactions were associated

with GHG emissions and do not allow us to infer causality. In future work, more

evidence from laboratory incubation experiments and functional gene analysis will be

needed to fully validate this hypothesis. The description has been added to the fourth

paragraph of revised section 4.3 *Extreme and normal patterns of carbon emissions*

*driven by picophytoplankton and heterotrophic bacteria* in Lines 624-628 of the clean

version of the revised manuscript. Specific revisions are below.

| **Revised manuscript** |
| --- |
| Extremely high $CH_4$ concentrations and fluxes mainly occurred in July (Fig. 2A), and extremely high values of $CH_4$ concentration were positively influenced by PP abundance ($p<0.001$; Fig. 7A). During the algal blooming period, high nutrient inputs and elevated temperatures facilitated the explosive growth of PP and HB in the warm season (Tang et al., 2017). The increased cell density of PP (Fig. S7) likely enhanced the possibility of interactions between PP and HB (Christie-Oleza et al., 2017). Increased PP-HB interactions facilitate cell aggregation, which increases carbon flux export to the bottom water column, thereby providing sufficient substrate for $CH_4$ production in the bottom layer (Gärdes et al., 2011). Additionally, eutrophication promotes positive interactions (i.e. cooperation) between PP and HB (Fig. S11), strengthening positive feedback between PP and HB (Coyte et al., 2015). These findings help explain why strong positive interaction strength (number of links) between phytoplankton and bacterioplankton was found |

in extreme carbon groups (eutrophic state) compared with normal groups (mesotrophic state) (Fig. 6). In eutrophic reservoirs, the efficient production and rapid decomposition of easily degradable autochthonous POC by PP and HB, respectively, accelerate $CH_4$ production in the short term (West et al., 2012; Gärdes et al., 2011), resulting in extremely high $CH_4$ concentrations and fluxes in the warm season (Fig. 2A). Indeed, a significant positive correlation between network degree (interaction strength) and $CH_4$ concentrations and fluxes was found under eutrophic conditions (Fig. S12). Such a positive correlation may support our third hypothesis; however, further validation through laboratory incubation experiments and functional gene analysis is still required.

---

## Author Comment (AC2)

**Response to Reviews for the manuscript #egusphere-2025-156**

We sincerely thank Referee #2 and the editor for reviewing and evaluating our manuscript entitled "Extreme carbon fluxes may result from autochthonous particulate organic carbon regulated by the interactions between picophytoplankton and heterotrophic bacteria in river-reservoir systems" (ID: egusphere-2025-156), and for providing us with valuable and constructive comments. These comments help us to improve the quality of the manuscript. We have revised the manuscript according to Referee #2's comments and provided detailed point-by-point responses to the comments. In the following, Referee #2's comments are shown in blue, and our responses are presented in black. The corresponding revisions have been highlighted in yellow in the revised manuscript.

Referee #2's key and minor comments and our responses are as follows:

Key comments included 15 points, which can be categorized into four aspects: (1) clarifying methodological details, (2) strengthening data interpretation, (3) refining statistical and ecological context, and (4) improving visualization and readability.

1) Aspect1 (extreme value classification): Provide goodness-of-fit metrics (e.g., Kolmogorov-Smirnov test) for the Pearson Type III distribution to validate its suitability for the dataset.

We thank you for pointing this out. We have used the Kolmogorov-Smirnov test to assess the differences between the measured data (i.e., $CH_4$ and $CO_2$ concentrations and fluxes of all samples) and theoretical distribution of Pearson type III frequency curve. The p-value from the Kolmogorov-Smirnov test has been added to Fig. S2, and

the descriptions regarding the use of the Kolmogorov-Smirnov test and fitness of the Pearson type III distribution have been included in section 2.5 of the revised manuscript and Supplementary Text S3 of the revised Supplementary Information.

| **Revised manuscript** |
|---|
| *2.5. Identification of extreme and normal levels of $CH_4$ and $CO_2$ concentrations and fluxes*

    Pearson type III probability distribution curve (Hosking and Wallis, 1997), a widely used probability distribution function in hydrology and meteorological statistics (Sun and Qin, 1989), such as frequency analysis of extreme hydrological events, risk assessment of extreme climate, etc. (Raynal Villaseñor, 2021). Here, we employed Pearson type III probability distribution to determine the extreme and normal values of $CH_4$ and $CO_2$ concentrations and fluxes. The $CH_4$ and $CO_2$ concentrations and fluxes of all samples fitted the Pearson type III distribution well (Fig. S1; Kolmogorov-Smirnov test, $p > 0.05$). |

| **Revised Supplementary Information** |
|---|
| **Text S3. Determination of extreme and normal values of greenhouse gases**
    Meanwhile, the sample data points and the theoretical Pearson type III curve representing the population were plotted on the Hessian probability grid paper (Fig. S1), which is the most widely applied probability distribution function in hydrology and meteorological statistics (Sun and Qin, 1989). The differences between our dataset and the theoretical Pearson type III distribution were evaluated using the Kolmogorov-Smirnov test, with the sample data fitting the theoretical curve well |

[Figure]

**Fig. S2. Pearson type III curve analysis on CH₄ (A, C) and CO₂ (B, D) concentrations and fluxes.** Ext_h, extremely high; Nor, normal; Ext_l, extremely low. CCH₄, and CCO₂ denote the concentration of CH₄, and CO₂ in the surface water column. FCH₄, and FCO₂ denote the flux of CH₄, and CO₂ across the water-air interface. The differences between measured data and the theoretical distribution of the Pearson type III frequency curve were evaluated using the Kolmogorov-Smirnov test, and *p* > 0.05 indicates that the dataset fits the theoretical curve well.

2) Aspect 1 (extreme value classification): Justify the 10% probability threshold for extreme values with ecological context (e.g., alignment with IPCC's "extreme event" definitions).

We thank you for pointing this out. According to the IPCC's definition, an extreme event (e.g., extreme weather or climate event) typically refers to an event with a low probability of occurrence that can induce sudden changes in ecosystems (IPCC, 2021). The IPCC generally uses a probability threshold to identify such extremes, with these events having a low probability, typically set at 10%. This means that events with a probability lower than 10% within a particular period and place are usually considered extreme (IPCC, 2021). Therefore, using a 10% probability threshold to define extremes is consistent with the ecological context and helps identify and study extreme events that have significant impacts on ecosystems. The rationale for using the 10% probability threshold in extreme identification has been added in section 2.5 of the revised manuscript. Specific revisions are below.

| Revised manuscript |
|---|
| *2.5. Identification of extreme and normal levels of $CH_4$ and $CO_2$ concentrations and fluxes* |
|  |
| Here, we employed Pearson type III probability distribution to determine the extreme and normal values of $CH_4$ and $CO_2$ concentrations and fluxes. The $CH_4$ and $CO_2$ concentrations and fluxes of all samples fitted the Pearson type III distribution well (Fig. S2; Kolmogorov-Smirnov test, $p > 0.05$). According to the IPCC's definition, an extreme event, e.g., an extreme weather or climate event, typically refers to an event with a low probability of occurrence that can induce sudden changes in ecosystems (IPCC, 2021). The IPCC generally uses a probability threshold to identify such extremes, with these events having a low probability, typically set at 10%. This means that events with a probability lower than 10% within a particular period and place are usually considered extreme (IPCC, 2021). Using a 10% probability threshold to define extremes is consistent with the ecological context and helps identify and study extreme events that have significant impacts on ecosystems. Therefore, in this study, we set the 10% probability ranges at the tails of both ends of the theoretical curve as the extreme values of $CH_4$ and $CO_2$ concentrations and fluxes (Ding and Jiang, 2009). Based on these threshold values, we divided the dataset into three groups, i.e. extremely high, normal, and extremely low. |

3) Aspect 1 (microbial functional analysis): Include flow cytometry gating strategies and quality controls (e.g., negative controls for FCM, rarefaction curves for sequencing).

We thank you for pointing this out. The description regarding the gating strategies and negative controls for flow cytometry has been added in the second and third paragraphs of Supplementary Text S1 of the revised Supplementary Information. Additionally, the rarefaction curves for sequencing have been added in Fig. S1. Specific revisions are below.

| Revised Supplementary Information |
|---|
| **Text S1. Flow cytometry analysis** |
| The fluorescence signals detected by the flow cytometer result from the combination of the cell's non-specific fluorescence and the specific fluorescence of the stains. In order to determine the non-specific fluorescence, we set up the negative controls. Before analyzing the samples, we first analyzed the negative control (i.e., samples without fluorescent stains added) and set the voltage for each channel (Chen and Cao, 2014). The positive/negative boundary was determined based on the non-specific fluorescence signal values. Then, the samples with fluorescent stains added were analyzed, and cells with fluorescence signal values above the boundary were identified as positive cells. |
| A total of 100,000 events were acquired for each sample. The cell concentration was computed by the sample flow rate and the the volume of sample plus additions (fixatives, beads, etc.) (Marie et al., 1999). Flow cytometer list mode files were collected, saved, and analyzed with CytoExpert software (Beckman Coulter® version 2.4, USA). The strategies for setting sorting gates can be summarized in three points: 1) exclude dead cells or fragments; 2) set the sorting gate at a relatively central |

position between the two cell populations to avoid overlap; 3) eliminate adherent

cells (Du and Feng, 2014). .

[Figure]

**Fig. S1. Rarefaction curves for bacterioplankton (A) and phytoplankton (B) based on observed ASVs.** Lines represent different samples

The results of rarefaction curves have been added in the second paragraph of

section 2.3 of the revised manuscript. Specific revisions are below.

| Revised manuscript |
| --- |
| The PCR products of each sample were sequenced on the Illumina MiSeq platform at Majorbio Bio-Pharm Technology, Co., Ltd. (Shanghai, China). A total of 3,400,170 and 1,763,782 high-quality 16S rRNA and 23S rRNA sequences were generated for 78 samples. After subsampling each sample to an equal sequencing depth (43,591 reads per sample for bacterioplankton and 22,613 reads per sample for phytoplankton), we clustered the reads into ASV tables, obtaining 7,855 ASVs for bacterioplankton and 2,106 for phytoplankton. The number of ASVs per sample ranged from 39 to 147 for bacterioplankton and from 47 to 466 for phytoplankton. |

> The rarefaction curves displayed clear asymptotes (Fig. S1), indicating near-complete sampling of the community. DNA extraction, PCR amplification, and high-throughput sequencing were discussed in detail in the supplementary method S2.

4) Aspect 1 (microbial functional analysis): Briefly describe primer sets and PCR conditions for 16S/23S rRNA sequencing to ensure reproducibility.

We thank you for pointing this out. We have added the description of the primer sets and PCR conditions in the second paragraph of section 2.3 of the revised manuscript. Specific revisions are below.

| **Revised manuscript** |
| --- |
| Genomic DNA was extracted in duplicate from the filters using the FastDNA SPIN kit (Mo Bio laboratories®, USA), following the manufacturer's instructions. The duplicate DNA extracts were mixed for the following polymerase chain reaction (PCR) amplification. For phytoplankton, the 23S rRNA gene was amplified using the primer pair A23SrVF2 (5'-CARAAAGACCCTATGMAGCT-3') and A23SrVR2 (5'-TCAGCCTGTTATCCCTAG-3') (Yoon et al., 2016). For bacterioplankton, the V3-V4 regions of the bacterial 16S rRNA gene were amplified using the primer pair 338F (5'-ACTCCTACGGGAGGCAGCAG-3') and 806R (5'-GGACTACNNGGGTATCTAAT-3') (Ding et al., 2020). Amplicons were purified with an AxyPrep DNA Gel Extraction Kit (Axygen Biosciences, USA). The PCR amplification was performed in a 20 μL reaction system containing 1 μL |

of DNA template, 0.8 μL of forward primer, 0.8 μL of reverse primer, 2 μL of 10×Buffer, 0.2 μL of BSA solution, and 2 μL of dNTPs (2.5 mM).

5) Aspect 2 (isotopic evidence): Discuss potential confounding effects of methanotrophy on $\delta^{13}$C-CH$_4$ values, particularly in hypoxic bottom waters. Clarify whether isotope measurements account for fractionation during gas exchange.

We thank you for pointing this out. In our study, we primarily focused on the variations in CH$_4$ and CO$_2$ concentrations and their isotopes in the water column. The processes of gas exchange at the water-air interface, such as dissolution and emission, were not considered and were assumed to cause negligible isotope fractionation. The potential effects of methane oxidation on CH$_4$ concentration and $\delta^{13}$C have been discussed in the second paragraph of section 4.1 of the revised manuscript. We also have added the distribution of $\delta^{13}$C-CH$_4$ in extremely normal and high groups of CH$_4$ concentration in Fig. S8. Specific revisions are below.

| Revised manuscript |
| --- |
| This is probably because the decomposition of OC from phytoplankton preferentially releases $^{12}$C and leaves the residual OC enriched in $^{13}$C (van Breugel et al., 2005). Furthermore, microbial oxidation of CH$_4$ can influence both its concentration and carbon isotope. Methane-oxidizing bacteria (MOB) are capable of consuming CH$_4$ under either anoxic or oxic conditions (Reis et al., 2020), leading to $^{13}$C enrichment of residual CH$_4$ (Whiticar and Faber, 1986). In freshwaters, the two main biological pathways of methanogenesis are acetate fermentation and CO$_2$/H$_2$ reduction (Preheim et al., 2016), both of which typically result in $\delta^{13}$C-CH$_4$ values |

lower than -50‰ (Whiticar, 1999) (acetate fermentation:-60‰ to -50‰; $CO_2/H_2$ reduction: -60‰ to -110‰). In our study, most $\delta^{13}C$-$CH_4$ values observed in the surface water column were higher than -50‰ (Fig. S8), suggesting that $CH_4$ was indeed oxidized in the five investigated reservoirs. The optimal DO concentration for $CH_4$ oxidation by MOB is reported to range from 0.5 to 4 mg/L (Thottathil et al., 2019). Here, in all five reservoirs, DO concentrations in surface water were generally above 4 mg/L. It can be inferred that $CH_4$ oxidation in the surface water is limited, and a considerable proportion of $CH_4$ is more likely oxidized in the bottom water or bottom sediment layer. High productivity may also convert lakes or reservoirs from a $CO_2$ source to a sink (Balmer and Downing, 2011).

[Figure]

**Fig. S8. Distribution of $\delta^{13}C$-$CH_4$ in extremely normal and high groups of $CH_4$**

**concentration.** The $\delta^{13}C$-$CH_4$ of methanogenesis by acetate fermentation or carbonate reduction is below the black doted line at -50‰ (Whiticar., 1999).

6) Aspect 2 (isotopic evidence): Expand on the paradox of elevated $\delta^{13}C$-DOC in high-Chl-a conditions (Fig. 8B): Does this reflect preferential $^{12}C$ uptake by PP or HB-mediated degradation?

We thank you for pointing this out. Picophytoplankton can preferentially absorb the isotopically light $CO_2$ ($^{12}C$) as a C source through primary production, leading to $^{13}C$ depletion in the surface water organic carbon (Han et al., 2018), and $\delta^{13}C$-DOC therefore exhibits a negative correlation with picophytoplankton. However, in summer, picophytoplankton assimilate $HCO_3^-$ as an alternative inorganic C source under $CO_2$ limitation during intense photosynthesis, which weakens the discrimination of $^{13}C$ and leads to $^{13}C$ enrichment of organic carbon (Fogel and Cifuentes, 1993). This can explain the significant positive linear correlation between $\delta 13C$-DOC and Chl-a in the extremely low group of $CO_2$ concentration (Fig. 8B). The explanation regarding the paradox of a significant positive correlation between $\delta^{13}C$-DOC and Chl-a has been added in the second paragraph of section 4.1 of the revised manuscript. Specific revisions are below.

| **Revised manuscript** |
| --- |
| Phytoplankton provides a large amount of autochthonous POC through photosynthesis, and picophytoplankton was therefore significantly correlated with POC ($p < 0.01$, Fig. 8A). Picophytoplankton can preferentially absorb the isotopically light $CO_2$ ($^{12}C$) as a C source through primary production, leading to $^{13}C$ depletion in the surface water organic carbon (Han et al., 2018), and $\delta^{13}C$-DOC |

therefore exhibits a negative correlation with picophytoplankton. However, in summer, picophytoplankton may assimilate $HCO_3^-$ as an inorganic C source under $CO_2$ limitation during intense photosynthesis, which weakens the discrimination of $^{13}C$ and enriches organic matter with $^{13}C$ (Fogel and Cifuentes, 1993). The above reasoning could explain the $^{13}C$ enrichment in the surface water DOC with the increase in Chl-a ($p < 0.01$, Fig. 8B).

7) Aspect 2 (PP-HB interaction mechanisms): Elaborate on how PP-HB cell aggregation (mentioned in Discussion) directly enhances $CH_4$ production. Does this relate to anoxic microenvironments or shared metabolic pathways (e.g., cross-feeding of vitamins)?

We thank you for pointing this out. The mechanism by which the aggregation of PP and HB cells enhances $CH_4$ production has been elaborated in the fourth paragraph of section 4.3 of the revised manuscript. Specific revisions are below.

| **Revised manuscript** |
| --- |
| Extremely high $CH_4$ concentrations and fluxes mainly occurred in July (Fig. 2A), and extremely high values of $CH_4$ concentration were positively influenced by PP abundance ($p < 0.001$; Fig. 7A). During summer algal bloom period, increased cell densities of PP and HB, induced by high nutrient inputs and elevated temperatures, would enhance the possibility of PP-HB interactions (Christie-Oleza et al., 2017). The intensified interactions, in turn, would promote the aggregation of their biomass. Increased aggregation of both PP and HB cells would increase carbon flux export to the bottom water column, providing sufficient substrates for |

CH$_4$ production in the bottom layer (Gärdes et al., 2011) and creating a micro-anaerobic environment that favors CH$_4$ production (Tang et al., 2023; Zhou et al., 2023).

8) Aspect 2 (PP-HB interaction mechanisms): Contrast freshwater PP-HB dynamics with marine systems (Introduction/Discussion), highlighting unique aspects (e.g., terrigenous POC influence).

We thank you for pointing this out. We have added a comparison of the patterns of PP-HB interactions between freshwater and marine systems in the fourth paragraph of the *Introduction* of the revised manuscript. Specific revisions are below.

| Revised manuscript |
|---|
| Marine and freshwater ecosystems exhibit distinct patterns of picophytoplankton-heterotrophic bacteria interactions: (i) picophytoplankton-regulated: In marine ecosystems, the interactions between picophytoplankton and heterotrophic bacteria appear to rely heavily on autochthonous POC produced by picophytoplankton through primary production (Morris et al., 2011); (ii) heterotrophic bacteria-regulated: In contrast, freshwater systems are closely linked to anthropogenic activities and act as collectors of large amounts of allochthonous POC from surrounding cities, towns, and farmlands (Wang et al., 2021). In these freshwater systems, picophytoplankton-derived autochthonous POC alone is insufficient to support heterotrophic bacterial growth, and the metabolic activity of heterotrophic bacteria primarily depends on allochthonous POC (Kritzberg et al., 2004). In addition, mineralization of |

allochthonous POC by heterotrophic bacteria releases nutrients and inorganic carbon, which, in turn, stimulate picophytoplankton growth (Solomon et al., 2015).

9) Aspect 3 (SEM path coefficients): Report standardized effect sizes (β values) for SEM paths to quantify the relative importance of PP vs. HB in driving extremes (Fig. 7).

We thank you for pointing this out. In Fig. 7, we have added the standardized total effects of each variable on extreme and normal values of $CH_4$ and $CO_2$ to quantify the relative importance of PP and HB in driving extremes. The description of the standardized total effects has been added in the last paragraph of section 3.5 of the revised manuscript. Specific revisions are below.

| Revised manuscript |
|---|
| In the extremely high group of $CH_4$ concentration, PP had the highest impact (0.90) on $CH_4$ concentration, followed by POCauto (0.42), POCallo (-0.31), and HB (-0.17) (Fig. 7F), based on the standardized total effects derived from the structural equation models depicted above. In the normal group, all variables had positive total effects on $CH_4$ concentration (Fig. 7G), with HB showing a slightly higher total effect than PP. In the extremely high group of $CO_2$ concentration, both PP and POCauto had positive total effects on $CO_2$ concentration (Fig. 7H). In the normal group of $CO_2$ concentration, only POCallo had a positive effect on $CO_2$ concentration (Fig. 7I). In the extremely low group of $CO_2$ concentration, POCauto had the highest impact on $CO_2$ concentration, followed by HB, PP, and POCallo (Fig. 7J). |

[Figure]

**Fig. 7.** Structural equation models (SEMs) describing selected variables' effects on the concentration of CH₄ (A, B) and CO₂ (C, D, E) in the extremely high (Ext_h), normal (Nor) and

extremely low (Ext_l) group, respectively. Numbers adjacent to arrows are standardized path coefficients and indicative of the effect size of the relationship. Solid arrows indicate significant paths (* $p < 0.05$, ** $p < 0.01$, *** $p < 0.001$), and dashed lines represent non-significant paths. The red and blue arrows indicate positive and negative path coefficients, respectively. The width of the arrows represents the strength of relationships. $R^2$ denotes the percentage of variance explained by the model. Standardized total (direct and indirect) effects of selected variables on the concentration of $CH_4$ (F, G) and $CO_2$ (H, I, J) in the extremely high (Ext_h), normal (Nor), and extremely low (Ext_l) group, respectively. PP, picophytoplankton; HB, heterotrophic bacterial; POCauto, autochthonous POC; POCallo, allochthonous POC; $CCH_4$ and $CCO_2$ were the concentrations of $CH_4$ and $CO_2$.

10) Aspect 3 (SEM path coefficients): Discuss why HB abundance showed no direct effect on $CH_4$ in SEM (Fig. 7A) despite high HB-PP correlations. Could functional genes (e.g., mcrA) resolve this discrepancy?

We thank you for pointing this out. In Fig. 7A, HB did not exhibit a significant direct effect on $CH_4$ concentration, despite strong HB-PP interactions. The possible reason is that, although HB are present, their potential for $CH_4$ production may vary depending on environmental conditions and microbial community activity. HB may influence $CH_4$ production at the functional level, which is not fully captured by HB abundance alone. Therefore, incorporating data on functional genes involved in methanogenesis, such as *mcrA*, could help clarify whether HB truly influence $CH_4$ production rather than relying solely on their abundance. The description has been added in the fourth paragraph of section 4.3 of the revised manuscript. Specific revisions are below.

| Revised manuscript |
| --- |
| Indeed, a significant positive correlation between network degree (i.e., |

interaction strength between HB and PP) and $CH_4$ concentrations and fluxes was found under eutrophic conditions (Fig. S14). Nevertheless, the SEM results indicated that HB did not exert a significant direct effect on extreme $CH_4$ concentrations under eutrophic conditions (Fig. 7A). A possible explanation is that HB may influence $CH_4$ production at the functional level, which is not fully reflected by HB abundance alone. Therefore, incorporating data on methanogenesis-related genes, such as methyl-coenzyme M reductase subunit A (*mcrA*), may help resolve this discrepancy. The strong positive correlation between PP-HB interactions and $CH_4$ concentrations and fluxes may support our third hypothesis; however, further validation through laboratory incubation experiments and functional gene analyses is still required.

11) Aspect 3 (Trophic State Gradients): Compare reservoir-specific results (e.g., Three Gorges vs. Xiaoba II) to assess how dam size/hydrology modulates POCauto contributions (Table S1).

We thank you for pointing this out. The comparison of dam-related and hydrological variables between TGR and XB II to assess how dam size or hydrology modulates autochthonous POC contributions has been added in the first paragraph of section 2.1 of the revised manuscript. Specific revisions are below.

| Revised manuscript |
| --- |
| XB II (with a total capacity of 11300 $m^3$) is a small reservoir only for drinking water supply located on a tertiary tributary named Ganxi Gulley of the Yangtze River. Compared to XB II, the TGR has a significantly larger dam size (total |

storage capacity and water depth) and a longer hydraulic retention time (HRT) (Table S1). The greater depth in TGR promotes thermal stratification, which inhibits vertical mixing and facilitates nutrient accumulation in the hypolimnion. Meanwhile, sufficient light in the epilimnion supports phytoplankton growth and increases the production of autochthonous POC. The extended HRT in TGR reduces light attenuation and prolongs the retention of nutrients and phytoplankton in the water column, enhancing primary production and further promoting the accumulation of autochthonous POC. In contrast, the smaller dam size and shorter HRT in XB II weaken stratification, reduce nutrient retention, and limit phytoplankton growth, resulting in a lower contribution of autochthonous POC. The geographical and project information of these selected reservoirs is described in Table S1.

12) Aspect 3 (Trophic State Gradients): Address seasonal biases: Why were extreme $CH_4$ fluxes concentrated in July (Fig. 2A)? Link this to thermal stratification or monsoon-driven nutrient pulses.

We thank you for pointing this out. We have discussed why extremely high $CH_4$ fluxes mainly occurred in July (Fig. 2A) in relation to the nutrient pulse in the fourth paragraph of section 4.3 of the revised manuscript. Specific revisions are below.

| **Revised manuscript** |
| --- |
| In eutrophic reservoirs, the efficient production and rapid decomposition of easily degradable autochthonous POC by PP and HB, respectively, accelerate $CH_4$ production in the short term (West et al., 2012; Gärdes et al., 2011; Benassi et al., |

2021; Beaulieu et al., 2019). During the July monsoon peak, thermal stratification is often disrupted, and nutrients accumulated in the hypolimnion are rapidly transported to the epilimnion—a process known as a nutrient pulse. This nutrient pulse enhances PP-HB interactions (Jung et al., 2016), accelerates the turnover of autochthonous POC, and thereby results in the extremely high $CH_4$ concentrations and fluxes observed in July (Fig. 2A). Indeed, a significant positive correlation between network degree (i.e., interaction strength between HB and PP) and $CH_4$ concentrations and fluxes was found under eutrophic conditions (Fig. S14).

13) Aspect 4: Figure 2B: Simplify boxplot labels (e.g., use "Ext_L," "Nor," "Ext_H") and adjust axis scales for NO3−-N/SRP to reduce overlap.

We thank you for pointing this out. We have replaced "Extremely low", "Normal", and "Extremely high" with "Ext_l", "Nor", and "Ext_h", respectively, in Fig. 2B. We also adjusted the scales of the vertical axis for environmental parameters, such as $NO_3^-$-N and SPR, in Fig. 2B to avoid overlap. The modified Fig. 2B is shown below.

[Figure]

**Fig. 2.** Comparison of months and environmental parameters in extreme and normal groups. Panel A Ternary plots showing the percentage of months (May, July, and November) in which the extremely low (Ext_l), normal (Nor), and extremely high (Ext_h) values of $CH_4$ and $CO_2$ concentrations and fluxes occurred. The yellow, blue, and red dots represent the Ext_l, Nor, and Ext_h groups. Panel B Characteristics of $NO_3^-$-N, SRP, DO, Chl-a, WT, pH, POC, and DOC in surface water. The yellow, blue, and red boxes represent environmental parameters in the extremely low (Ext_l), normal (Nor), and extremely high (Ext_h) groups of $CH_4$ and $CO_2$ concentrations and fluxes, respectively. Asterisks indicate significant difference: * $p < 0.05$, ** $p < 0.01$, *** $p < 0.001$.

14) Aspect 4: Figure 6: Add a legend for network edge colors (green = positive, violet = negative) and highlight keystone taxa (e.g., hub nodes) to emphasize PP-HB mutualism.

We thank you for pointing this out. We have identified the keystone taxa based on within-module connectivity (Zi) and among-module connectivity (Pi). The co-occurrence analysis, including the identification of keystone taxa, has been added

to Supplementary Text S6 of the revised Supplementary Information. Additionally, we have added the description of keystone taxa in the five networks in section 3.4 of the revised manuscript. We have also included a legend in Fig. 6 that illustrates edge colors and have labeled the keystone taxa using squares. Specific revisions are below.

| Revised Supplementary Information |
|---|
| **Text S6. Co-occurrence network**

The co-occurrence network (based on the ASV level) was constructed to disentangle potential interactions between bacterioplankton and phytoplankton. First, only amplicon sequence variants (ASVs) with a relative abundance > 0.1% of all samples were selected for correlation calculation (Xu et al., 2018). Pairwise correlation was then calculated based on the SparCC method, and only strong ($|r| > 0.6$) and significant ($p < 0.05$) correlations in the matrix were selected for further analysis (Hu et al., 2017). Networks were visualized in Gephi (v0.9.2). Network topological parameters of each sample, that is, subnetworks, were extracted using the subgraph function in the "igraph" package in R (Csardi and Nepusz, 2006). The nodes of each network were divided into four topological roles: (1) peripherals ($Zi \leq 2.5$, $Pi \leq 0.62$); (2) module hubs ($Zi > 2.5$, $Pi \leq 0.62$); (3) connectors ($Zi \leq 2.5$, $Pi > 0.62$); and (4) network hubs ($Zi > 2.5$, $Pi > 0.62$) (Olesen et al., 2007). Except for peripherals, the other three roles are considered keystone taxa (Banerjee et al., 2016). To identify keystone taxa, we calculated within-module connectivity (Zi) and among-module connectivity (Pi) using the "igraph" package in R (Fig. S4). |

*3.4. Interactions between phytoplankton and bacterioplankton communities*

The co-occurrence patterns of phytoplankton and bacterioplankton in the extreme and normal groups of $CH_4$ and $CO_2$ concentrations were determined using network analysis (Fig. 6; Table 3). Overall, the number of nodes ranged from 101 to 184 in all five interaction networks. Most networks consisted of more than 50% positive edges, except in networks for the normal group of $CO_2$ concentration. In addition, topological properties of the co-occurrence network in normal groups of $CH_4$ and $CO_2$, such as modularity, were higher than those in the extreme groups. In contrast, the average degree showed an exactly opposite trend. The number of phytoplankton-bacterioplankton links decreased from 977 in the extremely high group to 104 in the normal group of $CH_4$ concentration (Fig. 6A and B). Similarly, the number of phytoplankton-bacterioplankton links in the normal group was also significantly lower than both in the extremely high and extremely low groups of $CO_2$ concentration (Fig. 6C-E). Compared with communities in normal groups, communities in the extreme groups exhibited a higher interaction strength (that is, links) between phytoplankton and bacterioplankton. A total of 53 keystone taxa were identified from five networks (Fig. 6; Fig. S4), and the taxonomic information of these keystone taxa was listed in Table S4. These keystone taxa belong to bacterial clades including Proteobacteria, Actinobacteriota, and Bacteroidota, as well as phytoplankton clades such as Cyanobacteria and Bacillariophyta.

[Figure]

**Fig. 6.** Co-occurrence networks of phytoplankton and bacterioplankton communities based on correlation analysis. Panels A and B Co-occurrence patterns of phytoplankton-bacterioplankton interaction network in extremely high and normal groups of $CH_4$ concentration ($CCH_4$). Panels C, D, and E Co-occurrence patterns of phytoplankton-bacterioplankton interaction network in extremely high, normal, and extremely low groups of $CO_2$ concentration ($CCO_2$). Each line represents a significant correlation between the two taxa, with the green lines representing positive correlations and the violet lines representing negative correlations. The number of links represents the strength of interactions between phytoplankton and bacterioplankton. The red and blue nodes in each network represent phytoplankton and bacterioplankton, respectively. The size of each node is proportional to the number of connections (that is, degree). Circles and squares represent non-keystone taxa and keystone taxa, respectively. The taxonomic information of these keystone taxa was listed in Table S4.

15) Aspect 4: Supplement: Include a conceptual diagram summarizing the POCauto-PP-HB-$CH_4$/$CO_2$ pathway to guide readers.

We thank you for pointing this out. The conceptual summary of carbon ($CH_4$ and

$CO_2$) emission patterns under the influence of PP-HB interactions on POC from different sources is shown below.

[Figure]

Minor comments included 3 points.

1) Define abbreviations at first use (e.g., TLI in Table 2, SEM in Fig. 7).

We thank you for pointing this out. The full name of TLI (trophic level index) has been added to the caption of Table 2. In addition, the full name of SEM (structural equation model) has been included in the caption of Fig. 7.

2) Cite recent studies on reservoir eutrophication and $CH_4$ emissions.

We thank you for pointing this out. We have cited recent studies on reservoir eutrophication and $CH_4$ emissions in the fourth paragraph of section 4.3 of the revised manuscript. Specific revisions are below.

| Revised manuscript |
| --- |
| These findings help explain why strong positive interaction strength (number of links) between phytoplankton and bacterioplankton was found in extreme carbon groups (eutrophic state) compared with normal groups (mesotrophic state) (Fig. 6). |

> In eutrophic reservoirs, the efficient production and rapid decomposition of easily degradable autochthonous POC by PP and HB, respectively, accelerate $CH_4$ production in the short term (West et al., 2012; Gärdes et al., 2011; Benassi et al., 2021; Beaulieu et al., 2019).

3) Check consistency in decimal places (e.g., 0.07 vs. 0.070 mg·L−1 in Fig. S4).

We thank you for pointing this out. I have corrected the decimal places in the first paragraph of section 3.2 of the revised manuscript to ensure consistency throughout the paper. Specific revisions are below.

| **Revised manuscript** |
| --- |
| The $POC_{auto}$ and $POC_{allo}$ concentrations in whole dataset respectively ranged from 0.00 to 0.86 mg·L$^{-1}$ and 0.05 to 1.77 mg·L$^{-1}$, and were positively correlated with TLI. |